# Dissection of flag leaf metabolic shifts and their relationship with those occurring simultaneously in developing seed by application of non-targeted metabolomics

**Chaoyang Hu[1,2], Jun Rao[3], Yue Song[4], Shen-An Chan[4], Takayuki Tohge[5], Bo Cui[2], Hong Lin[2], Alisdair R. Fernie[5], Dabing Zhang[2], Jianxin Shi[2] ***

**1** Key Laboratory of Marine Biotechnology of Zhejiang Province, Key Laboratory of Applied Marine Biotechnology of Ministry of Education, School of Marine Sciences, Ningbo University, Ningbo, China, **2** Joint International Research Laboratory of Metabolic & Developmental Sciences, SJTU-University of Adelaide Joint Centre for Agriculture and Health, School of Life Sciences and Biotechnology, Shanghai Jiao Tong University, Shanghai, China, **3** Jiangxi Cancer Hospital, Nanchang, China, **4** Agilent Technologies Incorporated Company, Shanghai, China, **5** Central Metabolism Group, Max Planck Institute of Molecular Plant Physiology, Potsdam, Golm, Germany

* jianxin.shi@sjtu.edu.cn

**Data Availability Statement:** All relevant data are within the manuscript and its Supporting Information files.

## Abstract

Rice flag leaves are major source organs providing more than half of the nutrition needed for rice seed development. The dynamic metabolic changes in rice flag leaves and the detailed metabolic relationship between source and sink organs in rice, however, remain largely unknown. In this study, the metabolic changes of flag leaves in two *japonica* and two *indica* rice cultivars were investigated using non-targeted metabolomics approach. Principal component analysis (PCA) revealed that flag leaf metabolomes varied significantly depending on both species and developmental stage. Only a few of the metabolites in flag leaves displayed the same change pattern across the four tested cultivars along the process of seed development. Further association analysis found that levels of 45 metabolites in seeds that are associated with human nutrition and health correlated significantly with their levels in flag leaves. Comparison of metabolomics of flag leaves and seeds revealed that some flavonoids were specific or much higher in flag leaves while some lipid metabolites such as phospholipids were much higher in seeds. This reflected not only the function of the tissue specific metabolism but also the different physiological properties and metabolic adaptive features of these two tissues.

## Introduction

Rice (*Oryza sativa* L.) is the most important staple crop in the world feeding more than half of the world's population. With the current massive population growth coupled to near-global improvements of living standards, the pursuit for higher rice yield and equally high nutritive quality is ever increasing. Both the yield and quality of rice can be limited by the

**Funding:** This work was partly supported by grants from the National Natural Science Foundation for the Youth of China (Grant No. 31701400), the China National Transgenic Plant Special Fund (Grant No. 2016ZX08012-002), and the Program of Introducing Talents of Discipline to Universities (111 Project, B14016). This research was also sponsored by the K.C. Wong Magna Fund in Ningbo University. Yue Song and Shen-An Chan are employed by Agilent Technologies Incorporated Company. Agilent Technologies Incorporated Company provided support in the form of salaries for authors YS and SC, but did not have any additional role in the study design, data collection and analysis, decision to publish, or preparation of the manuscript. The specific roles of these authors are articulated in the 'author contributions' section.

**Competing interests:** We have the following interests: Yue Song and Shen-An Chan are employed by Agilent Technologies Incorporated Company. There are no patents, products in development or marketed products to declare. This does not alter our adherence to all the PLOS ONE policies on sharing data and materials.

supply of nutrition from leaves (source) and the ability to accumulate the available nutrition in seeds (sink) [1, 2]. The sink size (seed numbers and mean seed size) is established by seed organ morphogenesis and development during flowering and by source availability during grain filling, which is accompanied with the leaf senescence process [2]. During leaf senescence, catabolic processes such as the degradation of chlorophylls, proteins and lipids increase while anabolic processes decrease [3]. The shift from anabolic to catabolic metabolism in senescent leaves is vital for nutrition mobilization and recycling from leaf to developing seeds, which is important for seed yield and quality and has been studied mainly in Arabidopsis [3, 4]. The onset and progression of leaf senescence are often accompanied with the changes in gene expression [5, 6]. By using transcriptome analysis alongside molecular genetic techniques, many senescence-associated genes have been identified in rice [7–9]. Hundreds of differentially expressed proteins, involved in different cellular responses and metabolic processes, during senescence of flag leaves have additionally been investigated by proteomic analysis [10].

The flag leaf is the last growing leaf of rice. Its maximal growth rate in coincident to the time when the rice plant is at the heading and flowering stage and it becomes a mature leaf at around 7 days after flowering (DAF) [10, 11]. After fertilization, the leaves begin to continuously supply nutrients to the seeds for their growth and development, meanwhile, flag leaf senescence process gradually starts from about 14 DAF [10]. Rice seeds reach maturity at about 21 DAF [12], but the rice flag leaves do not wither even at 28 DAF for some cultivars. Given that it is present longer than the other leaves during the process of seed development and that it is physically closest to the developing seeds, the flag leaf can provide more than half of the nutrients needed for rice seed development [13]. Previous studies indicated that many of the quantitative trait loci (QTL) controlling flag leaf characteristics and yield-related traits are co-localized in rice [14, 15]. Therefore, the flag leaf is commonly regarded as the primary source of assimilates for yield and quality in rice.

It is well known that some of the small molecular compounds in the leaves are used only by the blades themselves, while others can be used both for the leaf itself and for the transport to the sink tissues/organs, for example, the developing seeds. The reported metabolic analysis of rice phloem sap revealed that many metabolites that synthesized in the leaves, including sugars (e.g. sucrose), amino acids (e.g. asparagine, glutamine and glutamate) and even flavonoids (e.g. tricin and schaftoside), can be transported through the phloem tubes to the sink tissues [16–18]. These results indicated an active metabolic flux from source to sink tissues. However, studies focusing on the metabolic changes of the flag leaves and their correlation with those of developing seeds across the seed development process in rice have not yet been elucidated.

The rice seed, as a sink, can not only store the nutritional metabolites transported from the leaves, but also can synthesize macromolecules such as starches, proteins and DNAs using small molecules transported from the leaves [19]. Nevertheless, it is not yet clear to what extent a specific metabolite in rice seeds is derived from the leaves or within the seeds themselves. Therefore, there is an urgent need to investigate the detailed metabolic shift from source to sink in rice, particularly across the seed development process. In a previous study, we revealed developmental stage and cultivar dependent metabolic changes in developing seeds of four rice cultivars [20]. It provided a comprehensive metabolic map in developing rice seeds [20].

In this study, we reported the metabolic shift in rice flag leaves from flowering to seed desiccation. We also investigated into the metabolic associations and differences in the levels of detected metabolites between flag leaves and developing seeds. The aim was to explore the possible metabolite association between source and sink organ in rice.

## Materials and methods

### Materials

Rice plants were planted in a paddy field in Minghang (31.03˚N, 121.45˚E), Shanghai, during the summer season in 2013. The tillers were marked at the heading dates. Flag leaves or rice seeds from two individual plants were pooled as one sample set (biological replication). Four sample sets (biological replications) of flag leaves at 0, 7, 14 and 28 days after flowering (DAF) and of rice seeds at 7, 14 and 28 DAF per rice variety were independently collected [20], immediately frozen with liquid nitrogen and lyophilized for 48 hours. Samples were ground into fine powder and stored at -80˚C until metabolomics analysis.

### Metabolite extraction, derivatisation and GC-MS analysis

Metabolite extraction: an aliquot of 10 mg fine powder was mixed with 700 μl 100% methanol containing an internal standard (0.01 mg/mL sorbitol), homogenize with tissue grinder (Jingxin, Shanghai, China) for 5 min at 30 Hz, followed by centrifugation at 14000 g for 10 min. An aliquot of 500 μl supernatant were taken into 2 ml fresh Eppendorf tubes, in which 300 μl chloroform was added. After gently mix, 750 μl pure water was added and the mixture was vortexed for 15 seconds. After phase separation, 150 μl upper phase was taken into 1.5 ml fresh Eppendorf tubes, dried exhaustedly in the speed vac for at least 3 hours without heating.

Metabolite derivatisation: An aliquot of 40 μl methoxyamine hydrochloride (20 mg/ml in pyridin) was added to the dried Eppendorf tubes containing the extracted metabolites, followed by shaking at 37˚C for 2 hours. Then 70 μl mixture of *N*-methyl-*N*-(trimethylsilyl)trifluoroacetamide (MSTFA) and fatty acid methyl esters (FAMEs) was added, and followed by shaking at 37˚C for 30 min. 100 μl of resulting solution was taken into sample vials for GC-MS analysis.

Data acquisition, metabolite identification and peak area extraction: The detailed GC-MS analysis was done as previously described [21]. The mass spectrometry data was sequentially collected in one batch, in which all samples were randomly ordered. The quality control (QC) samples equally pooled from all experimental samples were run after every 10 experimental samples. Metabolite identification and peak extraction were performed with TagFinder software [22].

### Metabolite extraction and LC-MS analysis

Metabolite extraction: an aliquot of 10 mg fine powder was mixed with 1 ml 100% methanol, and homogenized with tissue grinder (Jingxin, Shanghai, China) at 30 Hz for 5 min, and followed by centrifugation at 14000 g for 5 min. The supernatant was filtered through a syringe filter (0.22 μm) and placed in a sample vial for LC-MS analysis.

LC-MS data acquisition: The methanol extracts were analyzed by UHPLC-QTOF-MS in positive and negative mode, respectively. LC analysis was performed using an Agilent 1290 Infinity II LC™ system. MS detection was performed on an Agilent 6550 iFunnel/Q-TOF mass spectrometer with Agilent Jet-Stream source. The detailed UHPLC-QTOF-MS data acquisition was performed as described previously [20]. The samples were randomly ordered, and the QC samples equally pooled from all experimental samples were run after every 10 experimental samples.

Metabolite identification and peak area extraction: Metabolites were annotated by searching Personal Compound Database and Library (PCD/PCDL), literatures [21, 23, 24], and the Massbank [25] and Metlin [26] databases, based on two criteria: (1) the difference between the observed mass and the theoretical mass was less than 5 ppm; (2) the main feature of the

observed MS/MS spectrums was the same to that in literatures or database. Peak area extraction was performed with Mass Profinder 6.0 software. Each metabolite in every sample was carefully checked during peak area extraction to make sure that right peaks were extracted. The missing values were checked carefully with Mass Profinder software to make sure that missing values were caused by the content being too low to be detected not by random.

### Data analysis

IF a metabolite was detected simultaneously in GC-MS, UHPLC-MS negative mode and/or UHPLC-MS positive mode, the one with the smallest relative standard deviation (RSD) in the QC samples was retained. For data normalization of GC-MS data, peak area of a given metabolite was divided by the peak area of the internal standard (sorbitol), sample weight and the median value in all samples of the same metabolite. For data normalization of UHPLC-MS data, each set of 10 samples and each metabolite were normalized to the average level of QC samples that were injected before and after these 10 samples, then were divided by the sample weight and the median value in all samples of the same metabolite. The missing values of a given metabolite were imputed with the detected minimum value of the same metabolite in other samples for statistical analysis, assuming that they were below the limits of instrument detection sensitivity. The final statistics matrix with normalized data for the following statistical analysis are available in S1 Table.

Principal component analysis (PCA) was performed with SIMCA-P version 11.0 and the scaling type was "UV". Two-way ANOVA and ASCA were performed using the tool embedded in the MetaboAnalyst website (http://www.metaboanalyst.ca/) [27], by using "Log Normalization" for data transformation and "Autoscaling" for data normalization. Two-way ANOVA type used was "within subjects ANOVA", significance threshold was defined as the corrected $p$-value $< 0.05$ and False Discovery Rate (based on the Benjamini–Hochberg procedure) was chosen for multiple testing correction. ASCA was performed with default parameters. The metabolite-metabolite correlations between grain and flag leaf were analyzed by using Pearson's product-moment correlation method with mean values in R software package. The Student's t-test was employed to identify metabolites that differed significantly ($p < 0.05$) between two groups using the $log_2$ transformed data. The heatmaps of metabolite ratios were visualized with MultiExperiment Viewer (MeV) version 4.8 [28]. The figures were edited using Adobe Illustrator CS6 software for better resolution.

## Results

### Metabolite profiling of flag leaf samples

In order to investigate the metabolic changes of flag leaves along rice seed development, two *indica* subspecies (Qingfengai and 9311) and two *japonica* subspecies (Nipponbare and Nongken 58) were selected and planted in the same paddy field. Qingfengai and Nipponbare, having the same growth period (around 135 days after sowing), are medium maturing cultivars. Nongken 58 and 9311 with a growth period of about 160 days, are medium-to-late maturing cultivars. The flag leaf samples were collected at 0, 7, 14 and 28 DAF and extracted with methanol. The resulting methanol extracts were subjected to non-targeted metabolomics analysis by employing gas chromatography mass spectrometry (GC-MS) and ultra-high performance liquid chromatography-quadrupole time of flight-tandem mass spectrometry (UHPLC-Q-TOF-MS/MS). A total of 207 metabolites were identified (S2 Table), including 38 amino acids and dipeptides, 37 carbohydrates and organic acids, 25 lipids, eight nucleotides, 10 cofactors, five benzene derivatives, 63 flavonoids, 12 hydroxycinnamate derivatives, three terpenoids and six miscellaneous metabolites.

## Kinetic patterns of flag leaf metabolomes

To gain a global view of metabolic difference across all analyzed samples, principal component analysis (PCA) on the identified metabolites was subsequently performed. PC 1, accounting for 24.0% of the total variance, separated samples of *japonica* from those of *indica* (Fig 1), indicating different metabolic profiles in flag leaves of these two subspecies. This result was consistent with a previous report that the leaf metabolome at the five-leaf stage of *japonica* was significantly different from that of *indica* [29]. In either *indica* or *japonica* group, a cultivar dependent separation of samples of different sampling time points was also observed, with better separation in *indica* cultivars (Fig 1). This result indicated a developmental specific flag leaf metabolome, which was the same pattern as that of developing seeds [20].

Two-way ANOVA (Analysis of Variance) was conducted to decompose the raw data to further dissect which factor caused the variation of the metabolite levels. The abundances of 149, 183 and 142 metabolites were significantly affected by time, cultivar and their interaction, respectively. Among them, the abundances of 109 metabolites were simultaneously affected by time, cultivar and their interaction (Fig 2A). In addition, ANOVA-Simultaneous Component

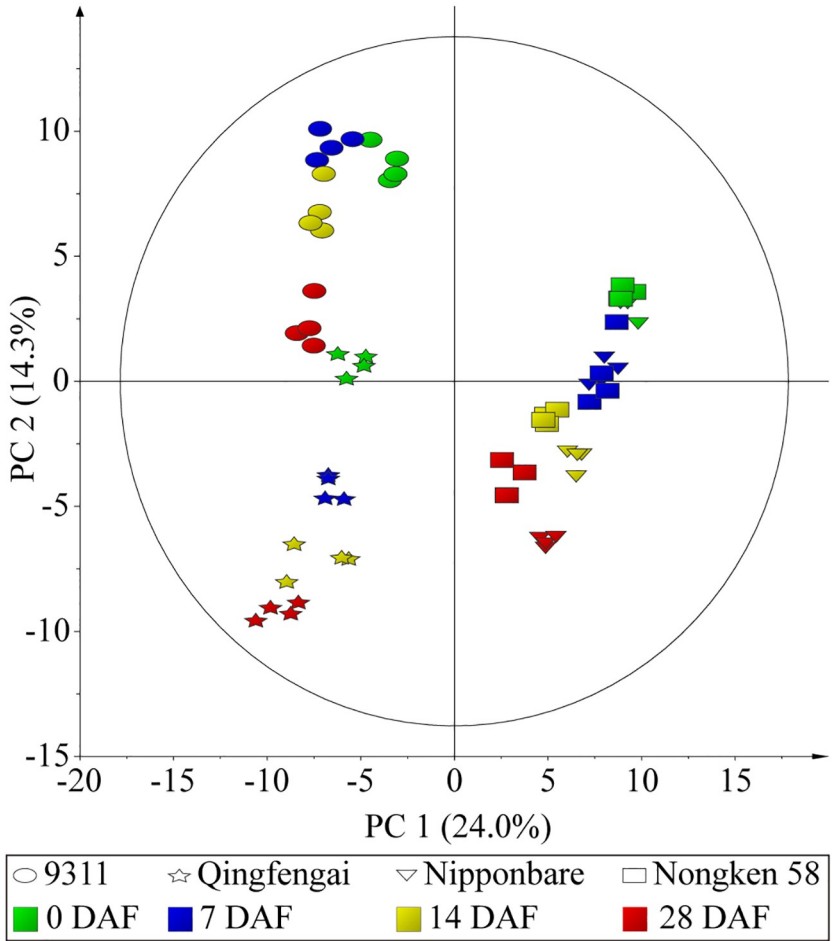

**Fig 1. Principal component analysis (PCA) of the metabolomes of flag leaves.** Green, blue, yellow and red colors represent samples at 0, 7, 14 and 28 DAF, respectively. Ellipse, star, triangle and square denote leaf metabolomes of 9311, Qingfengai, Nipponbare and Nongken 58, respectively. PC 1 explains 24.0% of variance distinguishing flag leaves of two *japonica* from those of two *indica*. PC 2 accounts for 14.3% of total variance separating leaf samples from different time points.

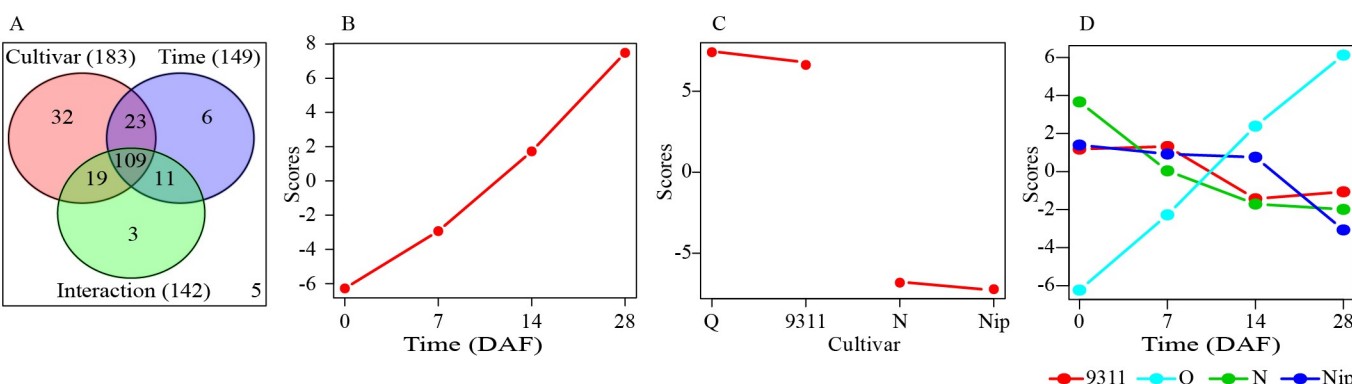

**Fig 2. Result of two-way ANOVA and ASCA.** (A) Venn diagram summary of results from two-way ANOVA. (B) Major pattern associated with cultivar. (C) Major pattern associated with time. (D) Major pattern associated with the interaction between cultivar and time. These analyses were performed in MetaboAnalyst website (http://www.metaboanalyst.ca/).

Analysis (ASCA), a multivariate extension of univariate ANOVA approach [30], was performed to identify the major patterns associated with each factor. The major pattern associated with time based on PC 1 of the corresponding sub-model was that the scores gradually increased from 0 DAF to 28 DAF and 73.86% of variation was explained by this sub-model (Fig 2B). Culture score plots showed that different cultivars differ in their PC1 scores; the scores of two *indica* cultivars were positive while those of two *japonica* cultivars were negative and 56.85% of variation was explained by this sub-model (Fig 2C). This result was consistent with PCA result shown in Fig 1, in which the samples of two *indica* cultivars located on the right side (the values of PC1 were positive) while those of two *japonica* cultivars on the left (the values of PC1 were negative). The first component of the interaction effect clearly showed the opposite trends occurring over the four sampling time points between Qingfengai and the other three cultivars and 25.32% of variation was explained by this sub-model (Fig 2D), which was different to that observed in developing rice seeds [20]. The observed difference between Qingfengai and other three cultivars reflected their physiological difference regarding senescence. At 35 DAF, the flag leaves of Qingfengai withered but those of other three cultivars did not, indicating an earlier or faster senescence of Qingfengai than others. The differences of the interaction scores among different cultivars at 7 and 14 DAF were smaller than those at 0 DAF and 28 DAF. This observed small and close interaction scores at 7 and 14 DAF across all four cultivars also resembled the conserved biochemical function of flag leaves at this specific grain-filling stage among tested cultivars. These three sub-models were validated using permutation test with the observed statistic *p* values all being less than 0.05 (S1 Fig).

Leverage/squared prediction error (SPE) plots were next made in order to identify metabolites that followed the major pattern of each factor [31]. Metabolites with high leverage and low SPE were picked out as well-modeled metabolites that contributed significantly to the model described above. Fourteen well-modeled metabolites, including 12 flavonoids, stood out based on the major pattern of cultivar (S2 Fig), which were important in distinguishing the leaves of two *japonica* from those of two *indica*. The levels of these 12 flavonoids were all significantly higher in *indica*, which was consistent with the observation that flavone mono-*C*-glycosides and malonylated flavonoid *O*-hexosides accumulated at higher levels in leaves of *indica* than in *japonica* [32]. Twenty well-modeled metabolites, including eight amino acids (asparagine, aspartate, glutamine, glutamate, gamma-guanidinobutyric acid, 5-oxoproline I, *N*-acetylglutamate and threonine), two carbohydrates (glycerate and glycolate), five cofactors

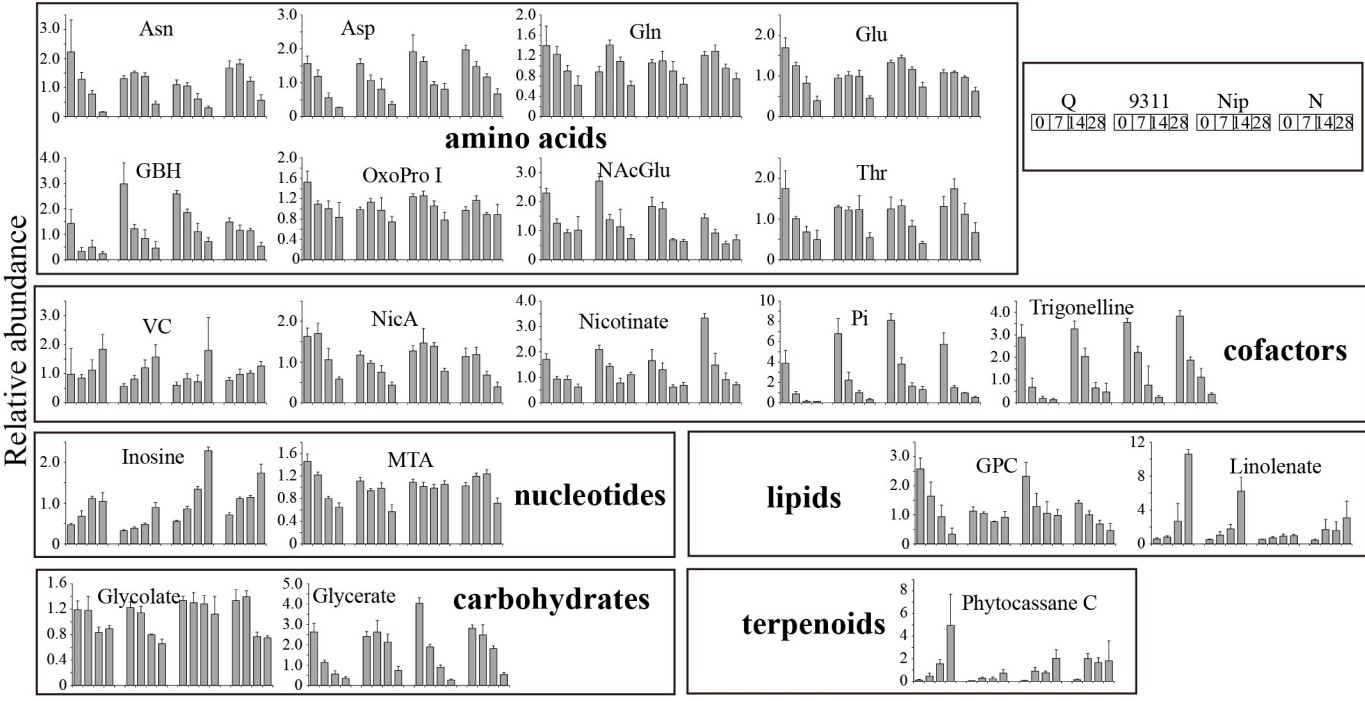

**Fig 3. Changes of the well-modeled metabolites following the major pattern of time.** For full metabolite names, refer to S2 Table.

(vitamin C, nicotinamide, nicotinate, phosphoric acid and trigonelline), two lipids (linolenate and glycero-3-phosphocholine), two nucleotides (inosine and 5-methylthioadenosine) and one terpenoids (phytocassane C), stood out based on the pattern of time (Fig 3). The levels of vitamin C, inosine, linolenate and phytocassane C gradually increased while those of the other 17 metabolites gradually decreased across time in all four cultivars. By contrast, 24 well-modeled metabolites followed the major pattern of interactive effect, which showed different change patterns along time between different cultivars (S3 Fig). Most of them were primary metabolites, including six amino acids (glycine, histidine, isoleucine, putrescine, tryptophan and valine), five carbohydrates (galactinol, raffinose, ribulose-5-phosphate, ribitol and threitol), seven lipids (1-LysoPE(18:3), 2-LysoPC(18:2), 9,13-DHOME, 9-HOTrE isomer, glycerol, linoleate and palmitate) and three nucleotides (adenosine, 5-methylthioadenosine and uridine) (S3 Fig).

## Metabolic changes during flag leaf maturation

To investigate the metabolic changes of flag leaves during maturation, the metabolite levels of leaves in a given cultivar at 7 DAF were compared with those at 0 DAF in leaves of the same cultivar. The levels of 66, 67, 78 and 98 metabolites were significantly changed in Qingfengai, 9311, Nipponbare and Nongken 58, respectively (S3 Table). The levels of methionine, gamma-guanidinobutyric acid, phosphoric acid and trigonelline were simultaneously decreased while those phenylalanine, isoorientin-*C*-hexoside derivant, 2-LysoPC(18:3), linolenic acid, inosine and phytocassane C increased in all four cultivars (Table 1, pattern A). The levels of seven flavonoids, including six tricin derivants and those of the other seven metabolites, such as VB$_2$, were significantly decreased and increased, respectively, in two *japonica* cultivars but not significantly changed in two *indica* cultivars (Table 1, pattern B). The levels of six metabolites,

**Table 1. Partial list of significantly changed metabolites during flag leaf maturation.**

| Metabolite Name[a] | Class | Ratio[b] | | | | Pattern[c] |
|---|---|---|---|---|---|---|
| | | Qingfengai | 9311 | Nipponbare | Nongken 58 | |
| Met | Amino acids | **0.45** | **0.43** | **0.56** | **0.68** | A |
| GBH | Amino acids | **0.23** | **0.41** | **0.72** | **0.78** | A |
| Pi | Cofactors | **0.23** | **0.33** | **0.47** | **0.26** | A |
| Meth-Nic | Cofactors | **0.24** | **0.63** | **0.63** | **0.49** | A |
| Phe | Amino acids | **1.82** | **1.62** | **1.42** | **1.64** | A |
| Isoo *C*-hex der | Flavonoids | **1.72** | **1.28** | **1.49** | **1.36** | A |
| 2-LysoPC(18:3) | Lipids | **1.45** | **6.39** | **2.10** | **1.37** | A |
| Linolenate | Lipids | **1.40** | **2.05** | **1.35** | **3.71** | A |
| Inosine | Nucleotides | **1.45** | **1.18** | **1.56** | **1.56** | A |
| Phytocassane C | Terpenoids | **4.29** | **4.99** | **13.99** | **14.74** | A |
| Tri der V | Flavonoids | 1.19 | 0.95 | **0.81** | **0.75** | B |
| Tri-4'*O*-ery-gua 7"*O*-glu | Flavonoids | 1.00 | 0.95 | **0.84** | **0.76** | B |
| Api-6*C*-glu-8*C*-ara II | Flavonoids | 2.47 | 1.24 | **0.72** | **0.77** | B |
| Tri-*C*-glu II | Flavonoids | 0.91 | 0.95 | **0.74** | **0.77** | B |
| Tri-4'*O*-ery-gua 7*O*-glu | Flavonoids | 0.97 | 1.01 | **0.86** | **0.78** | B |
| Tri-4'*O*-thr-gua | Flavonoids | 0.77 | 0.91 | **0.79** | **0.82** | B |
| Tri-4'*O*-thr-4-hyd | Flavonoids | 0.77 | 0.88 | **0.79** | **0.83** | B |
| Tricin isomer | Flavonoids | 0.95 | 1.00 | **1.36** | **1.15** | B |
| Isoscoparin | Flavonoids | 1.00 | 0.75 | **1.46** | **1.27** | B |
| Pro-*O*-hex II | Benzene derivatives | 0.32 | 1.56 | **1.75** | **1.45** | B |
| Lys | Amino acids | 1.03 | 1.07 | **1.55** | **1.87** | B |
| Glu-leu | Dipeptides | 0.73 | 1.25 | **2.33** | **2.18** | B |
| Uridine | Nucleotides | 0.53 | 1.69 | **1.77** | **2.82** | B |
| VB2 | Cofactors | 1.37 | 1.20 | **7.19** | **4.83** | B |
| DHA | Cofactors | 0.98 | **1.55** | 1.38 | **1.37** | C |
| His | Amino acids | 0.56 | **1.91** | 1.26 | **1.68** | C |
| Glucose | Carbohydrates | 0.84 | **4.19** | 0.96 | **2.29** | C |
| Fru | Carbohydrates | 0.87 | **3.30** | 0.97 | **2.35** | C |
| Val | Amino acids | 0.83 | **1.27** | 1.41 | **2.41** | C |
| 9-HOTrE | Lipids | 1.16 | **1.87** | 1.28 | **3.90** | C |
| Tri-4'*O*-ery-gua | Flavonoids | **0.77** | 0.91 | **0.80** | 0.84 | D |
| Glycerate | Carbohydrates | **0.43** | 1.09 | **0.47** | 0.88 | D |
| myo-Inositol | Carbohydrates | **0.40** | 1.03 | **0.71** | 1.02 | D |
| Maltose | Carbohydrates | **0.52** | 0.62 | **0.55** | 0.79 | D |

[a] Full names of the metabolites refer to S2 Table.

[b] Ratios of relative metabolite levels between 7 DAF and 0 DAF of the same cultivar. The bold values represent significantly different metabolic levels between 7 DAF and 0 DAF samples ($p$-values < 0.05). The $p$-values are available in S3 Table.

[c] The column of Pattern shows the metabolite change pattern during flag leaf maturation. A, the levels of metabolites were simultaneously increased or decreased in four cultivars; B, the levels of metabolites were simultaneously increased or decreased in two *japonica* cultivar; C and D, the levels of metabolites were simultaneously changed in two medium maturing cultivars and two medium-late maturing cultivars, respectively.

such as glucose and fructose, were significantly increased in two medium-late maturing cultivars (Table 1, pattern C) while those of four metabolites, such as glycerate and maltose, were significantly decreased in two medium maturing cultivars (Table 1, pattern D). More metabolic change patterns during flag leaf maturation were shown in S3 Table.

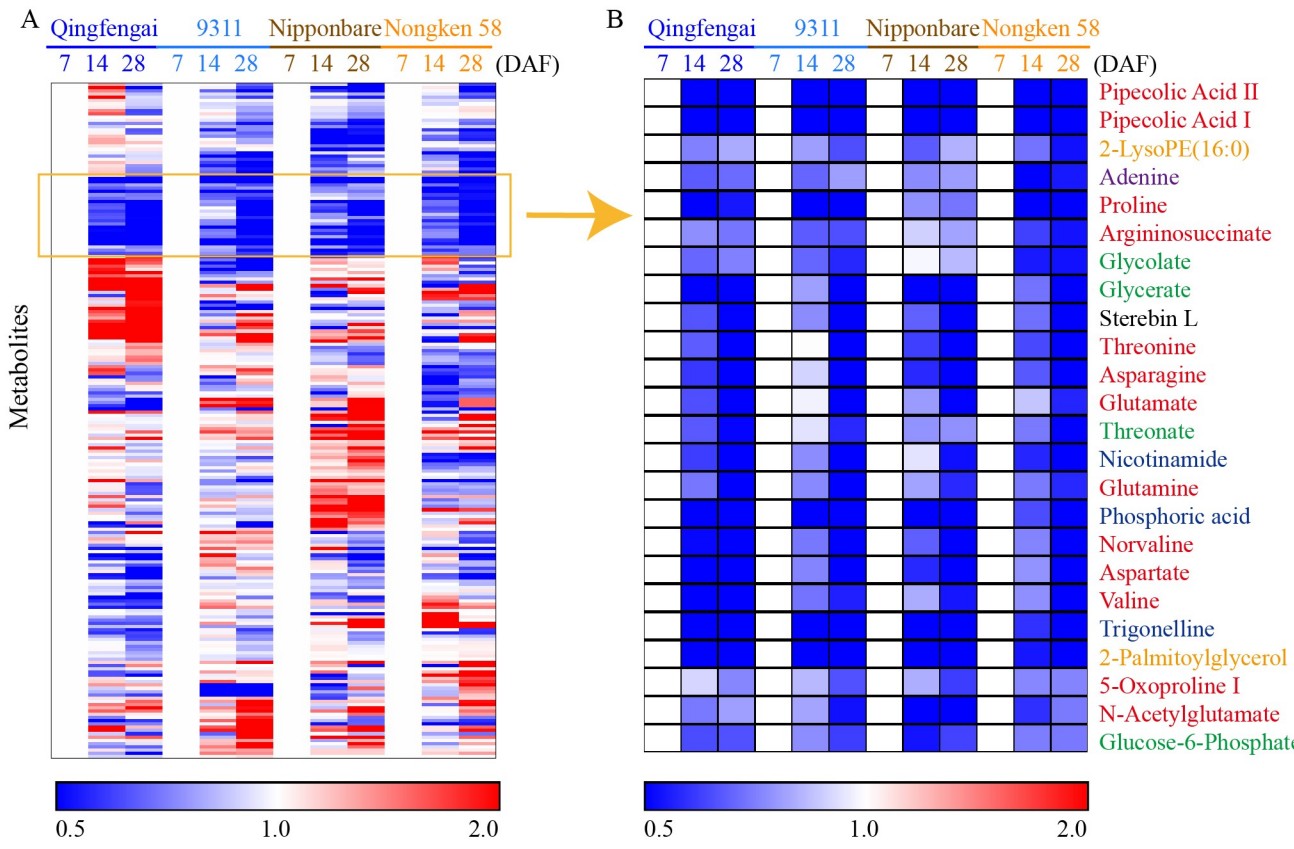

**Fig 4. Heat map of metabolite changes in flag leaves from 7 DAF to 28 DAF.** Q, Nip and N represent Qingfengai, Nipponbare and Nongken 58, respectively. Ratios of fold changes are given by shades of red or blue colors according to the scale bar. Data represent mean values of four biological replicates for each cultivar and time point. For full metabolite names, refer to S2 Table.

## Metabolic changes during flag leaf senescence

To uncover the metabolic alterations in flag leave senescence, the levels of the identified metabolites at 14 DAF and 28 DAF were compared with those at 7 DAF of the same cultivar to eliminate the cultivar-dependent variation and submitted to clustering analysis (S4 Table). The extent of metabolite changes during flag leaf senescence was much smaller than those in developing rice seed of the same time point as revealed in previous study [20]. Most of the metabolite change patterns varied among the four cultivars (Fig 4A). The levels of 24 metabolites were simultaneously decreased in all four cultivars during flag leaf senescence (Fig 4B). Thirteen of them were amino acids, including aspartate, asparagine, glutamate and glutamine. These amino acids are mainly synthesized in roots and shoots and serve as the major nitrogen storage and transport compounds of most non-leguminous plants [33]. This result implied a likely increased transportation of free amino acids in flag leaves to developing seeds during flag leaf senescence. The other ten of them were four carbohydrates (glycolate, glycerate, threonate and glucose-6-phosphate), three cofactors (nicotinamide, phosphoric acid and trigonelline), two lipids (2-LysoPE(16:0) and 2-palmitoylglycerol), and one single nucleotide (adenine). The remarkable consistency of these metabolic changes observed in all four cultivars (Fig 4B) suggested similar metabolism pattern of these metabolites in the senescence process.

**Table 2. Metabolites detected only in rice flag leaf or seed.**

| Metabolite Name | Formula | Class | Tissue |
|---|---|---|---|
| Apigenin-6-*C*-β-glucoside-8-*C*-α-arabinoside II | C26H28O14 | Flavonoid | Leaf |
| Chrysoeriol *C*-hexoside derivant | C25H28O12 | Flavonoid | Leaf |
| Chrysoeriol *O*-glucoside | C22H22O11 | Flavonoid | Leaf |
| Isoorientin-7,2''-di-*O*-glucoside | C33H40O21 | Flavonoid | Leaf |
| Isoorientin *C*-hexoside-C-hexoside II | C36H36O18 | Flavonoid | Leaf |
| Isoorientin | C21H20O11 | Flavonoid | Leaf |
| Isovitexin 2''-*O*-(6''''-(E)- feruloyl)-glucopyranoside | C37H38O18 | Flavonoid | Leaf |
| Syringetin 3-*O*-β-D-glucopyranoside | C23H24O13 | Flavonoid | Leaf |
| Tricin derivant I | C36H36O18 | Flavonoid | Leaf |
| Tricin derivant III | C36H36O18 | Flavonoid | Leaf |
| Tricin derivant IV | C36H36O18 | Flavonoid | Leaf |
| Tricin derivant V | C36H36O18 | Flavonoid | Leaf |
| Tricin 4'-*O*-(erythro-β-guaiacylglyceryl) ether 7-*O*-β-D-glucopyranoside | C33H36O16 | Flavonoid | Leaf |
| Tricin 4'-*O*-(syringyl alcohol)ether *O*-hexoside | C32H34O15 | Flavonoid | Leaf |
| Tricin 4'-*O*-(threo-β-4-hydroxyphenylglyceryl) ether | C26H24O10 | Flavonoid | Leaf |
| Tricin 4'-*O*-(threo-β-syringylglyceryl) ether 7''-*O*-β-D-glucopyranoside | C34H38O17 | Flavonoid | Leaf |
| Tricin 7-*O*-(2''-*O*-β-D-glucopyranosyl)-β-D- glucuronopyranoside | C29H34O17 | Flavonoid | Leaf |
| Tricin isomer | C17H14O7 | Flavonoid | Leaf |
| Tricin *O*-glucoside *O*-guaiacylglyceryl ether | C33H36O16 | Flavonoid | Leaf |
| Tricin *O*-guaiacylglyceryl ether'-*O*-glucopyranoside derivant | C36H36O18 | Flavonoid | Leaf |
| Tricin-*O*-hexoside derivative | C33H34O15 | Flavonoid | Leaf |
| 1-*O*-Palmitoylhexitol | C27H44O7 | Lipid | Leaf |
| Pregna-5,20-dien-3-ol | C21H32O | Others | Leaf |
| 1-linoleoylglycerol | C21H38O4 | Lipid | Seed |
| 2-linoleoylglycerol | C21H38O4 | Lipid | Seed |
| 1-myristoylglycerophosphocholine | C22H46NO7P | Lipid | Seed |
| 2-myristoylglycerophosphocholine | C22H46NO7P | Lipid | Seed |
| 1-stearoylglycerophosphoethanolamine | C23H48NO7P | Lipid | Seed |
| 2-stearoylglycerophosphoethanolamine | C23H48NO7P | Lipid | Seed |
| Cystine | C6H12N2O4S3 | Amino acid | Seed |
| Homovanillic acid | C9H10O4 | Monoaromatics | Seed |

## Metabolic difference between flag leaf and seed

In one of our previous studies, the metabolic profiles of developing seeds (at 7, 14 and 28 DAF) of the same four rice cultivars used in this study were characterized [20]. The sampling, metabolite extraction, and data acquisition of flag leaves in this study were all performed once together with these seed samples previous published [20], which facilitated our comparative study on metabolic difference between these two organs.

The most contrasting metabolic difference between the flag leaves and developing seeds as observed from the LC-MS chromatograph was the accumulation of flavonoids in flag leaves but lipids in seeds (S4A Fig). The closely gathered QC samples in the PCA score plot of flag leaves (S4B Fig) indicated a highly consistent and reliable analytical capacity of the system. The PCA score plot clearly demonstrated that flag leaf metabolome differed significantly from that of seed (S4B Fig). These results were further confirmed by the identification and quantification of those metabolites. Twenty three metabolites including 21 flavonoids were only detectable in flag leaves but not in seeds (Table 2). Half of the 23 metabolites were tricin or tricin glycosides,

which were active agents involved in plant defense system against bacteria, fungi and insects [34–36]. In contrast, eight metabolites including six lipids were detectable only in seeds but not in flag leaves (Table 2).

To explore the difference in metabolite abundance between flag leaves and seeds, the levels of metabolites in seeds were compared with those in flag leaves of the same cultivar at the same sampling time point (the ratios and *p*-values were presented in S5 Table), then the average of the ratios (AOR) of each metabolite was calculated. An AOR over 1 indicated a higher level of the metabolite in seeds, while an AOR less than 1 indicated a higher level of the metabolite in flag leaves. By doing so, the differences in abundance of a given metabolite between these two tissues could be easily identified. For example, a total of 29 metabolites were readily picked out with the AOR less than 0.01 (i.e. the levels of these metabolites in flag leaves were 100 times more than those in seeds; Table 3). Twenty five of these 29 metabolites were also flavonoids.

In addition, a total of ten metabolites displayed AOR values of more than 100 (i.e. the relative abundance of these metabolites in seeds was 100 times more than those in flag leaves). Eight of these ten metabolites were phospholipids and the other two of them were *N*-feruloyl-putrescine II and uracil (Table 3). Other metabolites that were found to be highly accumulated in seeds included certain lipids (such as LysoPE(16:0), LysoPE(18:2), linoleate, punicate and 9,13-DHOME), hydroxycinnamate derivatives (such as *N*-feruloylputrescine) and nucleotides (uridine, adenine and succinyladenosine) (S5 Table).

## Metabolic relationship between rice flag leaf and seed

It is interesting to investigate the metabolic association between flag leaves and seeds in order to better understand the metabolic relationship between source and sink organs. To this end, a pair-wise correlation analysis on metabolites identified in flag leaves of this study and metabolites identified in seeds of a previous study [20] was performed by employing the Pearson's product-moment correlation.

A total of 43,255 pairs of correlations were generated. Among them, there were 1245 pairs of positive correlations and 325 pairs of negative correlations with a threshold of absolute correlation value greater than 0.70 (|r-value| ≥ 0.70) (S6 Table), accounting for only 3.63% of the total possible correlations. Most of the positive correlations were observed between primary metabolites (amino acids, carbohydrates, cofactors, lipids and nucleotides) and hydroxycinnamates in seeds and primary metabolites in flag leaves (Fig 5). Most of the correlation coefficients (*r*-values) between secondary metabolites in seeds and those in flag leaves were quite small. However, some metabolite pairs of secondary metabolites with high correlation coefficients between flag leaves and seeds were also observed. For example, the levels of luteolin 7-*O*-glucoside, isoorientin 7,3'-dimethyl ether, chrysoeriol *C*-glucoside, chrysoeriol 6-*C*-β-gluco-side-8-*C*-α-arabinoside, and chrysoeriol -5-*O*-hexoside in seeds were highly correlated with that of isoorientin in flag leaves (r-value > 0.90, *p*-value < 1.5E-06) (S6 Table). This result suggested that isoorientin may be one of the secondary metabolites being transported via long-distance from flag leaves to developing seeds.

We investigated closely into metabolites whose levels showing corresponding changes between flag leaves and seeds. It generated a list of total 45 metabolites, including 15 amino acids, ten carbon rice compounds, seven flavonoids and four cofactors, whose levels were significantly correlated between seeds and flag leaves (Table 4). The levels of some amino acids, such as GABA, threonine, aspartate, valine, glycine, proline, glutamine, norvaline, alanine and glutamate in seeds correlated positively with those in flag leaves (*r*-values were 0.62~0.84). While the levels of other amino acids, including asparagine, betaine, gamma-guanidinobutyic acid, metonine and tryptophan correlated negatively with those in flag leaves (*r*-values were

**Table 3. The list of metabolites with AOR more than 100 or less than 0.01.**

| Metabolite [a] | Class [b] | Ratio [c] | | | | | | | | | | | | AOR [d] |
| | | Qingfengai | | | 9311 | | | Nipponbare | | | Nongken 58 | | | |
| | | 7 DAF | 14 DAF | 28 DAF | 7 DAF | 14 DAF | 28 DAF | 7 DAF | 14 DAF | 28 DAF | 7 DAF | 14 DAF | 28 DAF | |
|---|---|---|---|---|---|---|---|---|---|---|---|---|---|---|
| Tri-O-glu-O-gua der | Fla | 0.00020 | 0.00002 | 0.00002 | 0.00003 | 0.00005 | 0.00001 | 0.0001 | 0.00004 | 0.0001 | 0.00004 | 0.00003 | 0.00003 | **0.00005** |
| Tri-4'O-ery-gua 9''O-glu | Fla | 0.00026 | 0.00005 | 0.00021 | 0.00004 | 0.0002 | 0.00004 | 0.0001 | 0.00002 | 0.0001 | 0.00029 | 0.00005 | 0.00016 | **0.00012** |
| Isos-2''O-6'''-fer-glu | Fla | 0.00014 | 0.00019 | 0.00029 | 0.0001 | 0.00011 | 0.0001 | 0.0002 | 0.00019 | 0.0003 | 0.00042 | 0.00019 | 0.001 | **0.0002** |
| Tri-4'O-thr-gua 7O-glu | Fla | 0.00035 | 0.00016 | 0.00033 | 0.0001 | 0.00015 | 0.0001 | 0.0002 | 0.00027 | 0.0002 | 0.001 | 0.00016 | 0.00025 | **0.0002** |
| Tri-5O-glu | Fla | 0.001 | 0.00016 | 0.00017 | 0.0001 | 0.00011 | 0.0001 | 0.0002 | 0.00020 | 0.0002 | 0.00044 | 0.00025 | 0.00026 | **0.0002** |
| Isoo C-hex der | Fla | 0.00042 | 0.00016 | 0.00026 | 0.0002 | 0.0004 | 0.0002 | 0.0002 | 0.00013 | 0.0001 | 0.00037 | 0.00013 | 0.00020 | **0.0002** |
| Tri-4'O-ery-gua 7''O-glu | Fla | 0.00038 | 0.00026 | 0.0012 | 0.0002 | 0.001 | 0.0002 | 0.0001 | 0.00008 | 0.0001 | 0.00005 | 0.00005 | 0.00012 | **0.0003** |
| Isoo-2''O-glu | Fla | 0.001 | 0.001 | 0.001 | 0.0002 | 0.0003 | 0.0001 | 0.001 | 0.00029 | 0.0001 | 0.00047 | 0.00017 | 0.00016 | **0.0005** |
| Tri-4'O-ery-gua | Fla | 0.002 | 0.001 | 0.001 | 0.0003 | 0.001 | 0.0003 | 0.0004 | 0.00028 | 0.0004 | 0.00028 | 0.00020 | 0.001 | **0.0006** |
| Tri-4'O-thr-gua | Fla | 0.005 | 0.001 | 0.001 | 0.0003 | 0.001 | 0.0003 | 0.0003 | 0.00032 | 0.002 | 0.001 | 0.00021 | 0.001 | **0.0010** |
| Isoo-C-hex-C-hex I | Fla | 0.00049 | 0.00015 | 0.00032 | 0.003 | 0.006 | 0.002 | 0.0002 | 0.00009 | 0.0001 | 0.00038 | 0.00010 | 0.00013 | **0.0011** |
| Tri-7O-6'-mal-glu | Fla | 0.00007 | 0.00004 | 0.00006 | 0.00002 | 0.00006 | 0.00003 | 0.002 | 0.002 | 0.001 | 0.007 | 0.001 | 0.002 | **0.0012** |
| Isos-2''O-glu | Fla | 0.001 | 0.002 | 0.002 | 0.0003 | 0.0003 | 0.0002 | 0.003 | 0.001 | 0.001 | 0.005 | 0.001 | 0.002 | **0.0015** |
| Nar cha-C-pen-O-hex | Fla | 0.001 | 0.001 | 0.001 | 0.001 | 0.001 | 0.001 | 0.002 | 0.003 | 0.002 | 0.003 | 0.003 | 0.002 | **0.002** |
| Isocitrate II | Car | 0.003 | 0.001 | 0.003 | 0.007 | 0.003 | 0.001 | 0.001 | 0.001 | 0.0003 | 0.001 | 0.001 | 0.00040 | **0.002** |
| Tricin | Fla | 0.002 | 0.005 | 0.002 | 0.0002 | 0.001 | 0.0002 | 0.002 | 0.004 | 0.003 | 0.002 | 0.004 | 0.005 | **0.002** |
| Pro-O-hex I | Ben | 0.002 | 0.00047 | 0.00034 | 0.003 | 0.001 | 0.0001 | 0.008 | 0.002 | 0.002 | 0.010 | 0.003 | 0.001 | **0.003** |
| Isoo-7O-glu | Fla | 0.005 | 0.008 | 0.015 | 0.001 | 0.001 | 0.001 | 0.002 | 0.002 | 0.002 | 0.001 | 0.002 | 0.003 | **0.004** |
| Lut-7O-glu | Fla | 0.001 | 0.007 | 0.015 | 0.0002 | 0.001 | 0.001 | 0.001 | 0.002 | 0.002 | 0.005 | 0.006 | 0.004 | **0.004** |
| Lut-6C-2''O-glu-ara I | Fla | 0.003 | 0.005 | 0.005 | 0.002 | 0.003 | 0.0004 | 0.010 | 0.007 | 0.001 | 0.004 | 0.003 | 0.007 | **0.004** |
| Tri-C-glu I | Fla | 0.00046 | 0.003 | 0.004 | 0.0001 | 0.001 | 0.0005 | 0.003 | 0.012 | 0.005 | 0.001 | 0.013 | 0.012 | **0.004** |
| Tri-7O-6''-sin-glu | Fla | 0.004 | 0.009 | 0.011 | 0.005 | 0.004 | 0.004 | 0.001 | 0.004 | 0.006 | 0.002 | 0.002 | 0.006 | **0.005** |
| Lut-6C-2''O-glu-ara II | Fla | 0.004 | 0.012 | 0.002 | 0.001 | 0.0003 | 0.0003 | 0.011 | 0.009 | 0.001 | 0.011 | 0.002 | 0.005 | **0.005** |
| Isocitrate I | Car | 0.007 | 0.004 | 0.008 | 0.011 | 0.003 | 0.001 | 0.009 | 0.001 | 0.001 | 0.016 | 0.006 | 0.001 | **0.006** |
| 1O-Fer-glu II | Hyd | 0.003 | 0.002 | 0.002 | 0.008 | 0.001 | 0.0004 | 0.026 | 0.008 | 0.001 | 0.008 | 0.010 | 0.001 | **0.006** |
| Chr-6C-glu-8C-ara | Fla | 0.004 | 0.014 | 0.020 | 0.002 | 0.001 | 0.002 | 0.006 | 0.004 | 0.005 | 0.005 | 0.007 | 0.011 | **0.007** |
| Chr-C-hex-C-pen | Fla | 0.005 | 0.013 | 0.017 | 0.002 | 0.001 | 0.001 | 0.007 | 0.004 | 0.006 | 0.005 | 0.008 | 0.014 | **0.007** |
| Tri-C-glu II | Fla | 0.001 | 0.007 | 0.009 | 0.0005 | 0.001 | 0.002 | 0.004 | 0.018 | 0.016 | 0.001 | 0.010 | 0.014 | **0.007** |
| Tri-7O-glu | Fla | 0.001 | 0.002 | 0.003 | 0.001 | 0.001 | 0.001 | 0.004 | 0.033 | 0.011 | 0.004 | 0.036 | 0.015 | **0.009** |
| Fer-Put II | Hyd | 382.3 | 110.9 | 15.1 | 60.9 | 185.5 | 40.1 | 70.2 | 108.9 | 6.5 | 177.4 | 161.7 | 7.6 | **110.59** |
| 1-LysoPE(18:0) | Lip | 223.2 | 93.9 | 108.7 | 94.0 | 116.3 | 133.6 | 78.1 | 108.8 | 91.6 | 75.5 | 146.3 | 117.7 | **115.64** |
| 1-LysoPC(18:2) | Lip | 266.3 | 48.7 | 36.8 | 35.4 | 49.0 | 107.1 | 247.3 | 148.3 | 94.6 | 198.8 | 103.6 | 96.3 | **119.34** |
| 2-LysoPC(18:1) | Lip | 215.7 | 21.6 | 16.5 | 21.3 | 36.1 | 42.4 | 346.5 | 159.7 | 90.8 | 325.1 | 192.4 | 107.3 | **131.29** |
| 1-LysoPC(16:0) | Lip | 173.7 | 150.8 | 107.0 | 145.3 | 177.6 | 155.1 | 117.4 | 146.6 | 103.5 | 125.8 | 191.9 | 166.1 | **146.73** |

*(Continued)*

**Table 3.** (Continued)

| Metabolite [a] | Class [b] | Ratio [c] | | | | | | | | | | | | AOR [d] |
| | | Qingfengai | | | 9311 | | | Nipponbare | | | Nongken 58 | | | |
| | | 7 DAF | 14 DAF | 28 DAF | 7 DAF | 14 DAF | 28 DAF | 7 DAF | 14 DAF | 28 DAF | 7 DAF | 14 DAF | 28 DAF | |
| Uracil | Nuc | 210.5 | 33.0 | 1.1 | 482.7 | 172.7 | 1.8 | 406.5 | 103.5 | 1.6 | 408.9 | 741.3 | 3.6 | **213.92** |
| 2-LysoPC(18:2) | Lip | 310.2 | 270.8 | 215.6 | 99.7 | 173.0 | 327.8 | 190.9 | 313.9 | 232.9 | 229.9 | 263.8 | 439.8 | **255.69** |
| 2-LysoPC(16:0) | Lip | 531.0 | 340.6 | 266.7 | 435.2 | 584.2 | 657.3 | 518.6 | 869.1 | 398.7 | 326.4 | 515.6 | 488.0 | **494.28** |
| 1-LysoPC(18:1) | Lip | 431.9 | 359.4 | 245.7 | 604.4 | 693.3 | 593.3 | 658.3 | 1123.7 | 415.5 | 385.2 | 1100.9 | 442.2 | **587.80** |
| 2-LysoPC(14:0) | Lip | 580.3 | 1256.8 | 1036.3 | 220.1 | 1100.4 | 1121.9 | 241.1 | 1280.9 | 1466.6 | 116.8 | 348.3 | 1059.8 | **819.10** |

[a] Full metabolite names refer to S2 Table.

[b] Ben, benzene derivatives; Car, carbohydrates; Fla, flavonoids; Hyd, hydroxycinnamate derivants; Lip, lipids; Nuc, nucleotides.

[c] Ratio means the average level of metabolite in seed divided by that in leaf of the same cultivar and the same time point.

[d] AOR, the average of the ratios.

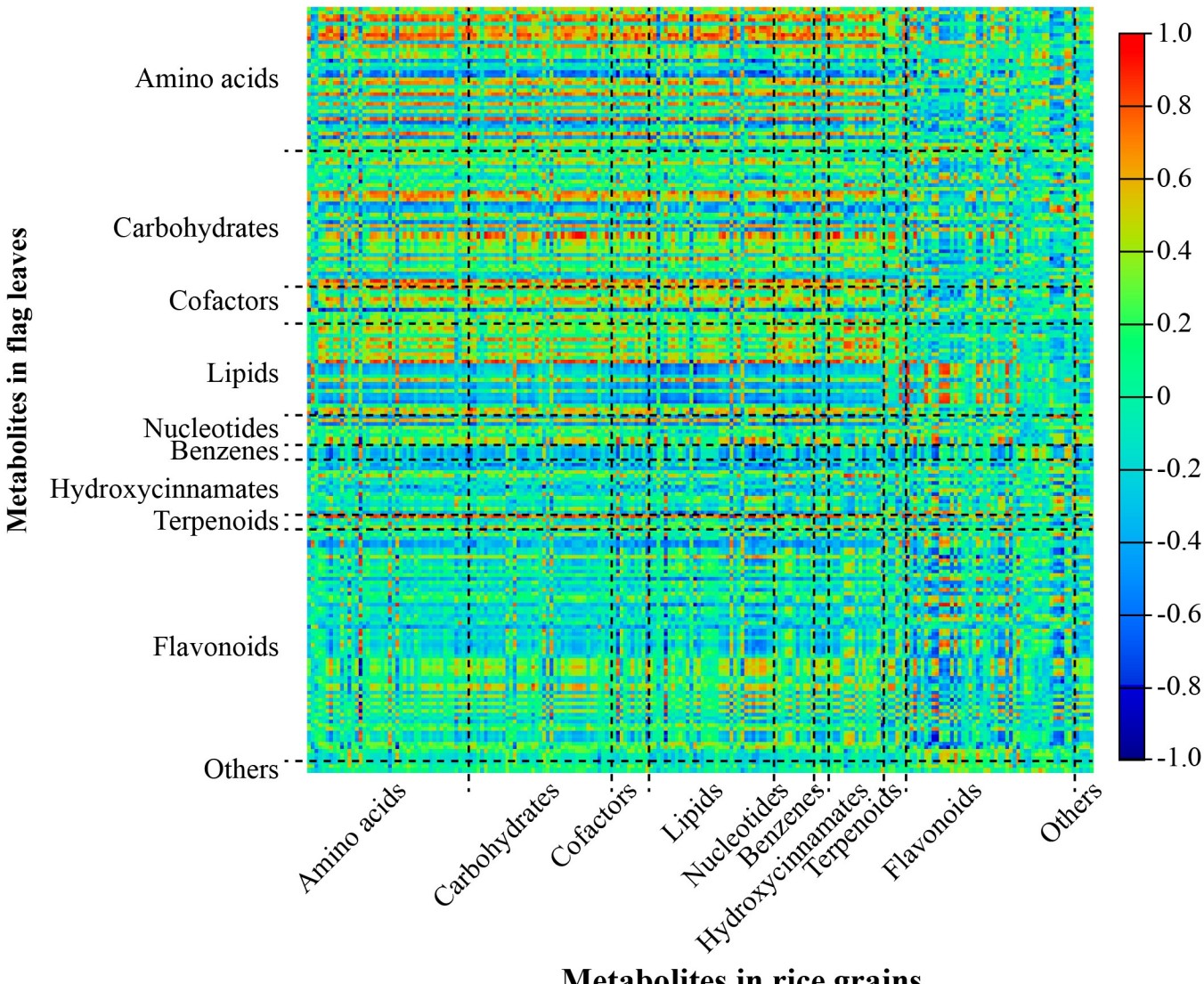

**Fig 5. Heatmap of metabolite-metabolite correlation between developing rice seeds and flag leaves.** Rectangles represent Pearson correlation coefficient ($r$) values of metabolite pairs (see correlation color key).

-0.63~-0.78). The levels of some carbon rich compounds such as pipecolic acid, maltose, glycerate, succinate, malate, myo-inositol, indole-3-carbaldehyde and 4-hydroxybutyric acid in seeds were positively associated with those in flag leaves. Pipecolic acid, a common lysine catabolite in plants and a critical regulator of plant defense system against pathogen [37], displayed the highest correlations ($r$-value > 0.96) between seeds and flag leaves. The levels of some cofactors, such as riboflavin, trigonelline, nicotinate and phosphoric acid, levels of some flavonoids, including tricin 7-$O$-neohesperidoside, swertisin, isoorientin 7,3'-dimethyl ether and chrysoeriol $C$-glucoside, and the level of phytocassane C, in seeds however were also positively correlated with those in flag leaves, with $r$-values of 0.65 ~ 0.72, 0.77 ~ 0.86, and 0.71, respectively. While the levels of three flavonoids, namely naringenin chalcone-$C$-pentoside-$O$-hexoside, isoorientin 7-$O$-glucoside and chrysoeriol-$C$- hexoside-$C$-pentoside and that of

**Table 4. Significantly associated metabolites between flag leaf and seed.**

| Metabolite Name | Class | r-Value | p-Value |
|---|---|---|---|
| GABA | Amino acids | 0.84 | 0.0006 |
| Threonine | Amino acids | 0.81 | 0.0015 |
| Aspartate | Amino acids | 0.80 | 0.0018 |
| Valine | Amino acids | 0.77 | 0.0035 |
| Glycine | Amino acids | 0.73 | 0.0065 |
| Proline | Amino acids | 0.72 | 0.0083 |
| Glutamine | Amino acids | 0.72 | 0.0085 |
| Norvaline | Amino acids | 0.69 | 0.0136 |
| Alanine | Amino acids | 0.66 | 0.0193 |
| Glutamate | Amino acids | 0.62 | 0.0329 |
| Tryptophan | Amino acids | -0.63 | 0.0284 |
| Methionine | Amino acids | -0.66 | 0.0206 |
| Gamma-Guanidinobutyric Acid | Amino acids | -0.66 | 0.0205 |
| Betaine | Amino acids | -0.68 | 0.0143 |
| Asparagine | Amino acids | -0.78 | 0.0026 |
| Benzoate | Benzenes | 0.58 | 0.0476 |
| Pipecolic Acid II | Carbohydrates | 0.98 | 3.74E-08 |
| Pipecolic Acid I | Carbohydrates | 0.96 | 1.32E-06 |
| Maltose | Carbohydrates | 0.72 | 0.0077 |
| Glycerate | Carbohydrates | 0.69 | 0.0132 |
| Succinate | Carbohydrates | 0.65 | 0.0223 |
| Isocitrate II | Carbohydrates | -0.63 | 0.0281 |
| Malate | Carbohydrates | 0.61 | 0.0339 |
| Myo-Inositol | Carbohydrates | 0.58 | 0.0466 |
| Indole-3-Carbaldehyde | Carbohydrates | 0.65 | 0.0232 |
| 4-Hydroxybutyric Acid | Cofactors | 0.62 | 0.0303 |
| Riboflavin | Cofactors | 0.65 | 0.0223 |
| Trigonelline | Cofactors | 0.72 | 0.0083 |
| Nicotinate | Cofactors | 0.71 | 0.0099 |
| Phosphoric acid | Cofactors | 0.65 | 0.0213 |
| Tricin 7-*O*-neohesperidoside | Flavonoids | 0.86 | 0.0003 |
| Swertisin | Flavonoids | 0.82 | 0.0010 |
| Isoorientin 7,3'-dimethyl ether | Flavonoids | 0.78 | 0.0027 |
| Chrysoeriol *C*-glucoside | Flavonoids | 0.77 | 0.0035 |
| Naringenin chalcone-C-pentoside-O-hexoside | Flavonoids | -0.67 | 0.0181 |
| Isoorientin 7-*O*-glucoside | Flavonoids | -0.62 | 0.0327 |
| Chrysoeriol-*C*-hexoside-*C*-pentoside | Flavonoids | -0.58 | 0.0461 |
| N-Feruloylputrescine I | Hydroxycinnamates | 0.87 | 0.0002 |
| 2-LysoPC(18:2) | Lipids | 0.65 | 0.0210 |
| 1-LysoPE(18:2) | Lipids | 0.59 | 0.0416 |
| 5-Methylthioadenosine | Nucleotides | 0.66 | 0.0193 |
| Succinyladenosine | Nucleotides | -0.58 | 0.0464 |
| Adenosine | Nucleotides | 0.58 | 0.0495 |
| Phytocassane C | Terpenoids | 0.71 | 0.0097 |

*r*-value, the correlation coefficient between flag leaf and seed.

succinyladenosine in seeds were negatively correlated with those in flag leaves, with *r*-values of -0.67 ~ -0.58.

## Discussion

The flag leaf of rice is the most important source organ for rice seed development. Therefore, understanding of the dynamic pattern of change in metabolites of the rice flag leaf and its metabolic relationship with the seed are of fundamental importance not only for the basic understanding of rice biology but also for applied rice breeding. In a previous study, we investigated the kinetic changes of the metabolomes of rice seeds at different developmental stages [20]. In the current study, we investigated the kinetic changes of the corresponding flag leaves from flowering to leaf senescence stage and the metabolic relationship between source and sink by taking advantage of an established non-targeted metabolomics platform.

### Conserved metabolic changes in flag leaf and seed

The patterns of dynamic changes in the metabolome of developing seeds are conserved among different rice cultivars [20], and the same is true for flag leaves as revealed by PCA in this study (Fig 1). Indeed both showed clear subspecies, cultivar, and stage dependent patterns, which indicated a conserved and well-coordinated metabolic regulation among source and sink tissues in rice. Nevertheless, the levels of the vast majority of metabolites in seeds gradually decreased across seed development especially at the seed desiccation stage in all four cultivars [20]. In contrast, most of the metabolites in flag leaves showed diverse patterns of change across the different cultivars. We postulate two possible explanations for this. First, rice seed is covered with a hard hull that provides a relative stable microenvironment for the growth and development of seed, perhaps rendering its metabolome less affected by the external environment [38]. While the flag leaf, being directly exposed to the changing natural environment, is more greatly influenced by environmental changes. Secondly, the development progression of different varieties of rice from fertilized eggs to mature seeds is very conservative [39]. Conservation of the pattern of dynamic change in the seed metabolome may reflect this robustness. However, the morphology and the structure of the flag leaf did not change significantly during rice seed development, yet the onset of leaf senescence varied among different rice cultivars. This may explain why the observed changes in the metabolic dynamics of the flag leaf were species specific.

### Metabolic difference between flag leaf and seed

It is well known that the quantity and chemical structure of plant metabolites are spatiotemporally distributed [40–42]. Although both metabolomes of flag leaves and developing seeds in rice showed similar kinetic patterns, clear metabolic differences were also observed. Twenty one flavonoids were identified only in flag leaves (Table 2), and the levels of 25 flavonoids and two phenolic acid-hexoside were much higher in the flag leaves than in the seeds (Table 3 and S5 Table). Flavonoids and phenolic acids (including protocatechuic acid and ferulic acid) are important in plants for scavenging reactive oxygen species (ROS) generated under excess light-stress [43, 44]. Flavonoids are also involved in plant defense against pathogens, pests, herbivores and UV irradiation [17, 43, 45, 46]. Rice seed is covered with a hard hull (palea and lemma), which also accumulates flavonoids and phenolic acids [44], and can protect seed from oxidative stress, predators and decay [47]. Therefore, the detected metabolic difference between flag leaves and developing seeds likely reflected different physiological properties and metabolic adaptive features of the two tissues. The secondary metabolites detected in flag leaves and developing seeds differed significantly (Tables 2 and 3). It is plausible to deduce that

these secondary protective metabolites detected in flag leaves and developing seeds were mainly synthesized and consumed locally. It was in agreement with the transcriptomic data reveled in flag leaves under heat stress, in which genes encoding rate-limiting enzyme in secondary metabolism were up-regulated [48]. Nevertheless, there are potential transport of flag leaf produced flavonoids (such as swertisin), and terpenoids (such as phytocassane C), to developing seeds as revealed in this study (Table 4).

It is also worth noting that some metabolites were found to be highly accumulated in seeds, particularly phospholipids and fatty acids. Lipids in rice seeds serve as stored energy and structural reserves, which can be converted into other metabolites when needed for seed development and subsequent germination [49]. Phospholipids and fatty acids play a considerable role in the seed germination capacity and protect seed from ageing [50, 51]. Therefore, the high accumulation of lipids observed in developing seeds but not flag leaves guaranteed proper seed development. The observed higher levels of some hydroxycinnamate derivatives in seeds implied an important role for them in the seed's defense response against biotic and abiotic stresses, owing to their antioxidant properties [52]. Whilst the high accumulation of feruloyl-putrescine, one of the large diversity of best characterized defense metabolites against insects and herbivores [53], which additionally serves as a storage form of polyamine, could be associated with germination potential in rice seeds [54]. The higher levels of uracil and uridine in seeds also suggested a unique biochemical basis for seed physiology, especially for seed starch biosynthesis. Arabidopsis *PYD1* (*dihydropyrimidine dehydrogenase 1*) knockout mutants accumulated high uracil levels in seeds and showed delayed germination [55]. While the accumulation of uridine in potato tubers was accompanied by increased amounts of starch in the tubers [56] and knockout of plastid uridine scavenging pathway resulted in reduced starch in Arabidopsis seed [57]. Further detailed investigations into those greatly different metabolites between seeds and flag leaves would likely expand our understanding of the biochemical mechanisms underpinning seed developmental process.

### Metabolic association between flag leaves and seeds

Nutrient remobilization and translocation from leaf to seed during leaf senescence is vital for the yield and quality in crops and as such has been subject to considerable research attention [58, 59]. The flag leaves of rice are the major source organs supplying developing seeds. Exploring the metabolite-metabolite correlations between flag leaves and seeds will facilitate the better understanding of the source-sink relationship in rice. Only a small proportion of metabolites were significantly correlated between flag leaves and developing seeds during rice seed development. Positive and negative relationships were found mainly in primary and secondary metabolites, respectively. These results indicated that metabolites in source tissues function not only as direct structural and nutritional molecules for development, but also as precursors for the synthesis of the secondary metabolites needed for defense in sink tissues. It is reported that, during rice grain filling stage, those genes involved in starch biosynthesis, lipid biosynthesis, seed storage protein synthesis, or genes encoding various transporters, such as sugar transporters, ABC transporters, amino acid/peptide transporters, phosphate transporters and nitrate transporters, are significantly up-regulated [60]. Therefore, more attention should be paid to increase the sink ability at seed filing stage to improve the nutrition of rice seeds in future breeding.

Water-soluble carbohydrates, such as sucrose, are the major non-structural storage carbohydrate fraction in leaves and stems of cereals [61]. It is well known that flag leaves contribute a significant amount of carbohydrates to the developing seeds [62]. However, in this study, the correlations of the levels of carbohydrates between flag leaves and seeds were not significant.

This was consistent with a previous proteomic study during senescence of flag leaves, proteins involved in carbohydrate metabolism was largely unchanged [63]. Another reason behind this could be that the carbohydrates transported from the flag leaves were rapidly transformed into other storage molecules by the developing seeds. In support of this notion, it is reported that flux of glycolysis and oxidative pentose phosphate pathway are very active in developing *Brassica napus* embryos [64]. However, similar studies has not been conducted in rice, which merits to be done in the future to fully address this issue.

It is noteworthy that above 20% (45 out of 208) of the metabolites in flag leaves were significantly correlated with the same metabolite in developing seeds (Table 4). The correlation of pipecolic acid, a non-proteinaceous product of lysine catabolism, between flag leaves and developing seeds was the highest. Pipcolic acid is reported to be involved in plant systemic acquired resistance (SAR) and basal immunity to bacterial pathogen infection, which accumulates in inoculated and distal leaf tissue in Arabidopsis [65–68]. We speculated that pipecolic acid in developing seeds may be mainly transported from rice flag leaves, because the seeds are covered by the hull while the leaves are exposed to an environment teaming with potentially harmful microbes. In support of this hypothesis is the observation that high amounts of pipecolic acid are transported from source to sink tissues through the sieve tube system in cucumber [69]. Ten amino acids, including two essential amino acids for human (valine and threonine) and three dominant nitrogen forms in the phloem (glutamine, glutamate and aspartate) [33, 70, 71], were positively associated between flag leaves and seeds (Table 4). This correlation indicated a likely direct source-sink transport of these metabolites. This suggest a potential application perspective for amino acid improvement in rice seeds could be achieved by elevating amino acid levels in rice leaves, increasing the export of them from leaves, or both, via manipulating related gene expression in leaves [72]. A transcriptomic study in rice flag leaves during rice grain filling indicated the up-regulation of many transporters for mineral and amino acid transportation from flag leaf to seed [73]. In addition, the proteomic data from aging rice flag leaves also reported the up-regulation of proteins in amino acid and glycolysis metabolism and in transportation [63]. However, the possibility that part of these metabolites come from roots can not be excluded currently.

Secondary metabolites are normally synthesized and accumulated in specific tissues or cell types for specific functions [74]. Most of the secondary metabolites are transported from source cells to neighboring cells, but some of them can also been transported to other tissues or remote organs [75, 76]. We found the levels of four flavonoids and two terpenoids in developing seeds were positively correlated with those in flag leaves (Table 4), indicating that these metabolites may indeed be synthesized in flag leaves and transported to developing seeds.

## Prospect for future research

Here we presented the kinetic metabolomes of flag leaves and associated it with our previously published metabolome of seeds along seed development across four different cultivars. This was done by taking advantage of exactly the same handling methods from sampling down to data acquisition and analysis in these two tissues. Although interesting correlation-ships of many metabolites between the two tissues were revealed and could be supported by known publications, essential additional work to address the metabolic flux between flag leaf to seed in rice needs to be done in the future using different methods and materials. To our knowledge, the best method to elucidate the direct metabolic flux could be done using traceable radioactive labeled representative key metabolites of different classes. For this purpose, targeted methods rather than non-targeted methods should be used.

## Conclusions

This study investigated the metabolite change patterns of rice flag leaves along seed development (from flowering to seed desiccation), and compared it with that in developing seeds reported previously [20], revealing both cultivar-, tissue- and development- dependent metabolic changes in rice. It furthermore revealed association of metabolic changes in flag leaves with those in seeds, providing important hints for us to better understand the metabolic relationship between source and sink tissues of rice. This observation, combined with future works additionally employing transcriptomics and proteomics techniques, will facilitate both the exploration of fundamental questions regarding the relationship between source and sink as well as their potential applications in rice seed nutrition improvement.

## Supporting information

**S1 Fig. The results of model validations through permutations.**
(DOCX)

**S2 Fig. Heatmap of the metabolites which stood out based on the major pattern of cultivar.**
(DOCX)

**S3 Fig. Heatmap of the 24 well-modeled metabolites followed the major pattern of interactive effect.**
(DOCX)

**S4 Fig. Different metabolome of rice flag leaf and developing seed.** (A) The LC-MS total ion chromatograph (positive ion mode) of flag leaf and developing seed of Qingfengai at 7 DAF. (B) PCA score plot of the metabolomes of rice flag leaves and developing seeds.
(DOCX)

**S1 Table. The final statistics matrix with normalized data.**
(XLSX)

**S2 Table. List of metabolites identified in flag leaves.**
(XLSX)

**S3 Table. Metabolite changes during flag leaf maturation.**
(XLSX)

**S4 Table. Metabolite changes during flag leaf senescence.**
(XLSX)

**S5 Table. Metabolic difference between rice seed and flag leaf.**
(XLSX)

**S6 Table. Data of metabolite-metabolite correlations analysis.**
(XLSX)

## Acknowledgments

We acknowledge the technical assistance of Shanghai Yuan Pu Biotechnology Co, ltd. for metabolomics analysis. We are also grateful to the four anonymous reviewers and editor for their constructive comments and suggestions for improving the manuscript.

## Author Contributions

**Conceptualization:** Chaoyang Hu, Jun Rao, Jianxin Shi.

**Data curation:** Chaoyang Hu, Jun Rao, Bo Cui, Hong Lin.

**Formal analysis:** Chaoyang Hu.

**Funding acquisition:** Chaoyang Hu, Jianxin Shi.

**Methodology:** Yue Song, Shen-An Chan.

**Supervision:** Dabing Zhang, Jianxin Shi.

**Visualization:** Chaoyang Hu, Hong Lin.

**Writing – original draft:** Chaoyang Hu.

**Writing – review & editing:** Chaoyang Hu, Takayuki Tohge, Alisdair R. Fernie, Jianxin Shi.

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
