## [Decision Letter · Decision Letter 0]

15 Nov 2019

PONE-D-19-26924

Dissection of flag leaf metabolic shifts and their relationship with those occurring simultaneously in developing seed by application of non-targeted metabolomics

PLOS ONE

Dear Dr. Shi,

Thank you for submitting your manuscript to PLOS ONE. After careful consideration, we feel that it has merit but does not fully meet PLOS ONE’s publication criteria as it currently stands. Therefore, we invite you to submit a revised version of the manuscript that addresses the points raised during the review process.

We would appreciate receiving your revised manuscript by Dec 30 2019 11:59PM. To enhance the reproducibility of your results, we recommend that if applicable you deposit your laboratory protocols in protocols.io, where a protocol can be assigned its own identifier (DOI) such that it can be cited independently in the future. For instructions see: http://journals.plos.org/plosone/s/submission-guidelines#loc-laboratory-protocols

We look forward to receiving your revised manuscript.

Kind regards,

Haitao Shi

Academic Editor

PLOS ONE

Journal Requirements:

1. 

3.  Thank you for stating the following in the Financial Disclosure section: "This work was partly supported by grants from the China National Transgenic Plant Special Fund (Grant No. 2016ZX08012-002, 2014ZX08012-002), the Program of Introducing Talents of Discipline to Universities (111 Project, B14016), and National Natural Science Foundation for the Youth of China (Grant No. 31701400).

We note that one or more of the authors are employed by a commercial company: Agilent Technology, Inc.

Reviewers' comments:

Reviewer's Responses to Questions

**Comments to the Author**

1. Is the manuscript technically sound, and do the data support the conclusions?

Reviewer #1: Partly

Reviewer #2: Yes

Reviewer #3: Yes

Reviewer #4: Yes

2. Has the statistical analysis been performed appropriately and rigorously? 

Reviewer #1: No

Reviewer #2: Yes

Reviewer #3: Yes

Reviewer #4: Yes

3. Have the authors made all data underlying the findings in their manuscript fully available?

Reviewer #1: No

Reviewer #2: Yes

Reviewer #3: Yes

Reviewer #4: Yes

4. Is the manuscript presented in an intelligible fashion and written in standard English?

Reviewer #1: Yes

Reviewer #2: Yes

Reviewer #3: Yes

Reviewer #4: No

5. Review Comments to the Author

Reviewer #1: The authors apply untargeted mass spectrometry-based metabolomics to compare metabolic differences between flag leaves and rice seeds during development. A number of technical issues need to be addressed:

1. The number of biological replicates (n=4) seems unnecessarily low, especially given the ready accessibility of rice leaves/seeds. The low number of replicates raises concerns about the statistical significance of the findings.

a. Further troubling is the following statement in the methods:

“Two flag leaves from two individual plant were pooled as one biological replication. Flag leaves or rice seeds from two

individual plants were pooled as one biological replication.”

Under most circumstances, there is no reason to pool biological replicates. This only masks the “true” biological

variance between replicates. Now, if the reason that multiple leaves/seeds were pooled is because a single leave/seed

did not provide enough sample for detection purposes, then this should have been discussed in the main text.

Importantly, this would have further justified the need for the number of biological replicates to be greater than four.

2. GC-MS and UHPLC-MS in the positive and negative mode were used to characterize the leaves/seeds metabolome. This is fine and actually a strength of the proposal, but what is not addressed is how the data sets were normalized to each other? The authors state that the data was normalized to sample weight. This is insufficient. There is too much instrument and sample preparation variability to strictly rely on only a simple constant-number normalization scheme.

a. What about internal/external standards?

b. What about QC samples? How was batch variability handled?

c. How were the samples collected? Was the sample order randomized? Were the GC-MS and UHPLC-MS collected

simultaneously? Sequentially? On different days? Was the same sample sub-sampled for GC-MS and UHPLC-MS

or were different samples prepared?

d. Could all of the observed variance be attributed to how the samples were prepared and handled, and how the

spectra were collected, instrument variation and insufficient normalization? Because of the nature of MS data, it

is “easy” to obtain distinct groups, but it can be difficult to validate that the differences are real.

3. On pages 8-10, the authors over-interpret the significance and the meaning of the relative/comparative meaning of scores/loadings from the various PCA models. Relative trends within a given model are fine, but there is not a point of reference to compare between PCA models. The appearance of a PCA scores plot can change dramatically from any number of reasons, such as changing the order of the samples in the data matrix, changing normalization/scaling methods, removing/adding samples, etc. Given that this section does not provide any real insight or contribute significantly to the study, I would recommend removing it. The fact that the metabolomes vary as a function of time and cultivar is sufficient.

4. In the data analysis section, the authors state: “False Discovery Rate is chosen for multiple testing correction.” But, it is not clear if all reported p-values are FDR corrected p-values or what type of FDR correction method is used.

5. The heatmap, Figure 4, should be plotted with all biological replicates (not average values) and with hierarchal clustering in both dimensions. This may further emphasize additional consistencies (like Figure 4B) or other trends across the entire dataset.

6. I understand the choice behind grouping the metabolites by class in the correlation map (Figure 6). But, as is, it fails in being informative. Instead, the metabolites should be ordered from the highest positive to the lowest negative correlation. Then, clusters of metabolites that belong to the same metabolic pathway, cellular process or chemical class within the highly positive/negative correlation should be labeled.

Reviewer #2: Hu and colleagues address here the understudied subject of the metabolic contribution of the flag leaf (a term that refers to the last growing leaf of rice that is key to the carbon flux into the grain). Flag leaf development is coincident with flowering, thus the properties of this leaf are tightly connected to the final outcome in terms of growth, yield and content of the rice grain. For this reason, for understanding the metabolics events taking place during rice seed development as well as sink-source relationship, it is key to study both flag leaf and grain. The study here used an extensive metabolic platform to investigate the metabolic dynamic changes of flag leaves and grain in 4 rice cultivars (2 indica and 2 japonica subspecies) and to give a broad insight into the influx of metabolites from flag leaf to grain, as well as into the outcome of grain in the light of natural variation. As the study is descriptive in nature, it is raising an important contribution to our current knowledge on this important subject and can path the way for future work: it will be very interesting to follow the secondary metabolites identified here as candidates to be transported from flag leaf to seeds, and their role .

The presentation of the work is easy to follow; data and experiments are well documented and in the data analysis appropriate statistical analysis was applied. I have few minor comments / suggestion:

Figure 5: in my opinion, this figure could be shifted to a supplementary file.

Page 28 lines 452: pipecolic acid has indeed emerged in recent years as a key defense mediator. I can see the possible role of this signal metabolites in grain for deterring pathogens. Could you also add here a sentence about the possible negative correlation (impact) between Pip and final grain weight as a result of the defense cost.

Reviewer #3: The study of Hu et al titled "Dissection of flag leaf metabolic shifts and their relationship with those occurring simultaneously in developing seed by application of non-targeted metabolomics" investigated the metabolic changes of flag leaves in two japonica and two indica rice cultivars using non-targeted metabolomics approach. This study revealed that flag leaf metabolomes varied significantly on both species and developmental stage, with only a few of the metabolites in flag leaves showing the same pattern of change in the four tested cultivars. The authors also found that levels of 45 metabolites in seeds that are associated with human nutrition and health correlated significantly with their levels in flag leaves.

This study is well organized and provides important insight into the nature of the metabolic flux from source to sink organs in rice.

Some minor comments are as follows:

Line 34, &135: Please use the uniform presentation of Principal component analysis (PCA) throughout the manuscript.

Line 47: “in the word” ?

Line 176-177: “relatively small, relative large”, I feel the usages are quite strange.

Line 292-293: Please rewrite this sentence.

Line 325: rice grain or rice seed? It would be better to keep it uniform throughout the manuscript.

Line 351: R-value or r value? It would be better to keep it uniform throughout the manuscript.

Line 354: the word “arguably” used here is quite confusing.

Line 446: “that fact that flux modelling in in”, a “in” should be deleted.

Line 504: “giving” should be “given”

Line 527: “these species” are ambiguous

Reviewer #4: Metabolic changes in flag leaves and developing seeds of four rice cultivars were analyzed. Limited overlap in metabolite features were observed among the four lines. Forty-five metabolites enriched in seeds showed a similar pattern of accumulation in leaves and seeds. The authors hypothesize that these data “revealed not only the function of the tissue-specific metabolites but also provide important insight into the nature of the metabolic flux from source to sink organs in rice”. Not certain if this was actuallydone, there were no flux measurements. Just levels at discrete timepoints, with the leaf samples analyzed much later than the seed samples. There are number of places in the manuscript with long run-on sentences with diverse ideas. These should be rewritten with simpler sentences.

Abstract:

Lines 40-44 needs to be rewritten, especially lines 42-44 which should be deleted. These statements have nothing to do with this study (they should be ok in the last section of the discussion).

Introduction:

Line 57 – brackets are missing for reference 3.

Lines 57-62 – Please rewrite, not sure what the authors are trying to convey here.

Line 102 – we report changes in metabolite levels in flag leaves and developing seed from flowering to seed dessication. The (our) aim was to reveal the source and sink metabolite relationships in rice.

Results:

Lines 145 to 147 – these varieties show similar senescence patterns (from Introduction), so it should not be surprising that their metabolomes were relatively similar. Also, for Lines 146 to 149.

Line 160 – Figure 2 C, the line joining the 2 sets of cultivars should be deleted. They need to describe a bit better why Qingfengai has a very different pattern than the other 3 cultivars. Not fully certain about the need for Figure 2. Figure 1 sums up their data and Figure 3 and later show differences if any among the 4 cultivars.

Line 237 – should be “extent”

Figure 3 – quite surprising that only linolenate increased with time in flag leaves (expected from lipid breakdown), and the overall patterns for the other metabolites very similar (all decrease over time).

Line 241-244 – Not sure what this sentence intends to convey? Would not changes in amino acids in flag leaves be largely driven by senescence between 14 and 28 days? Is it not more likely that proteins broken down to amino acids were being mobilized from the flag leaves to developing seeds? The remarkable consistency in data shown in Figure 4B would suggest similarities in the senescence process.

Discussion:

A considerable problem the authors face in interpretation is as follows:

By and large a bulk of the metabolites unloaded into developing seeds can be expected to be monomers, which are likely to be converted rapidly within developing seeds into polymers, such as proteins, starch, membranes, RNA, and DNA. That is flag leaf metabolism is dissimilatory during senescence, whereas seed metabolism is assimilatory prior to desiccation. The authors consider this point but should discuss the shortcomings of their current experimental protocol in somewhat greater detail.

A case in point is Table 4, where they detected significant associations in metabolites between seeds and leaves. However, it is difficult to discern if there was/is a direct biological correlation between these levels in the two tissues. Can they be sure these compounds came from the flag leaves to seeds, and say not from root transfer?

There are other aspects the authors should consider – both flag leaf senescence and seed development in rice have been intensely studied using physiological, biochemical and molecular tools. It may help if they could incorporate key findings from such studies within the scope of their metabolite analyses. They mention as much in their conclusions.

It might be best to revise the manuscript to stay within the key elements of their work (flag leaf metabolomes) and draw a simpler inference to their earlier work with seeds [ref 20]. It is also not clear if the instrument settings and detector sensitivity were the same for the two experiments. There was no mention of an internal standard used if it was they should add this to their M&M section.

Materials and Methods:

Lines 482-484 appear to be repeated?

I am not sure I fully understand their approach to metabolites missing in one or more samples. It might be best to exclude these metabolites from further analyses. It was either present or absent in a given extract. What exactly is the median value of a metabolite, area in all four samples, area in 2 out of 4 replicates? Please clarify. The authors could consider treating them as missing values and make comparisons based on samples or treatment groups with detectable levels. Another alternative would be to do a binomial test (e.g. Chi-square) based on presence or absence of a metabolite.

6. PLOS authors have the option to publish the peer review history of their article (what does this mean?). If published, this will include your full peer review and any attached files.

Reviewer #1: No

Reviewer #2: No

Reviewer #3: No

Reviewer #4: No

---

## [Author Response · Author response to Decision Letter 0]

19 Dec 2019

Response to reviewers' comments:

Reviewer #1: The authors apply untargeted mass spectrometry-based metabolomics to compare metabolic differences between flag leaves and rice seeds during development. A number of technical issues need to be addressed:

1. The number of biological replicates (n=4) seems unnecessarily low, especially given the ready accessibility of rice leaves/seeds. The low number of replicates raises concerns about the statistical significance of the findings.

Answer: Thank you for raising this good question. The reason why four not more biological replicates is based on followings: 1) Generally, three to six biological replicates are recommended for untargeted metabolomics (Vinayavekhin N and Saghatelian A, 2010, Current Protocols in Molecular Biology). 2) Our lab established good protocol for sample collection and preparation, which followed exactly Metabolon company’s SOP, and significantly reduced the effect of this process on metabolic change. Actually, the consistency of the sampling can be seen clearly in the PCA score plot of Fig. 1, in which four biological replicates of the same group are closely gathered together while those of different groups are well separated. 3) From practical point of view, if one more replicate is added, 40 more injects for GC-MS, UHPLC-MS positive and negative modes are required, respectively. This addition not only increases the workload, but also causes the incomplete analysis of all samples in one batch, increasing difficulty of data analysis. Therefore, four biological replicates for each sample is the chosen in our lab for effective metabolomics analysis. In past years, our lab carried out many metabolomics studies in plants and other systems, all obtained satisfactory results. 

Please refer to our publications

In Plant system: Plant Journal (Hu et al., 2016); Scientific Reports (Hu et al., 2014; 2016), Metabolomics (Rao et al., 2014), Journal of Integrative Plant Biology (Lin et al., 2014; Qu et al., 2014), Plant Cell Reports (Kim et al., 2017), International Journal of Molecular Science (Hu et al., 2018; Duan et al., 2019; Hu et al., 2019), and Algal Research (Hu et al., 2019).

In other system: World Mycotoxin Journal (Liu et al., 2016); Chemico-Biological Interactions (Liu et al., 2017). 

2. Further troubling is the following statement in the methods:

“Two flag leaves from two individual plant were pooled as one biological replication. Flag leaves or rice seeds from two individual plants were pooled as one biological replication.” Under most circumstances, there is no reason to pool biological replicates. This only masks the “true” biological variance between replicates. Now, if the reason that multiple leaves/seeds were pooled is because a single leave/seed did not provide enough sample for detection purposes, then this should have been discussed in the main text. Importantly, this would have further justified the need for the number of biological replicates to be greater than four.

Answer: This is again a good question, which we would like to clarify particularly for omic studies except genomics.

Because the nature of the metabolomics, which is a snapshot of a specific tissue /cell at specific stages under specific environment condition, therefore, many factors affect significantly the signature of the metabolomics status. As such, sample subpooling is a strategy used to reduce the variance but still allow studies to encompass biological variation. Underlying sample pooling strategies is the biological averaging assumption that the measurements taken on the pool are equal to the average of the measurements taken on the individuals (Natasha A. Karp and Kathryn S. Lilley, 2009, Proteomics). 

It is widely accepted in metabolomics study. For example, in “Precautions for harvest, sampling, storage, and transport of crop plant metabolomics samples”, supersamples of each variety are independently prepared to ensure representativeness of the physiological variations of a given variety of fruit at a certain development stage by harvesting several fruits of each variety to constitute a sample set (Biais B, et al., 2011, Plant Metabolomics).

In our case, we pooled two flag leaves from two individual plants as a sample set and four samples sets were independently prepared and analyzed. 

To make it clear, we have rephrased “biological replication” as “sample set (biological replication)” in the M&M section of the revised manuscript.

3. GC-MS and UHPLC-MS in the positive and negative mode were used to characterize the leaves/seeds metabolome. This is fine and actually a strength of the proposal, but what is not addressed is how the data sets were normalized to each other? The authors state that the data was normalized to sample weight. This is insufficient. There is too much instrument and sample preparation variability to strictly rely on only a simple constant-number normalization scheme.

Answer: Thank you for raising this important issue. 

When raw data is not good enough (for example, caused by the instability of the machine), data normalization with internal/external standards or with QC samples is required. 

In our analysis, all samples (including leaf, seed, and QC samples) were extracted at the same time, analyzed by the same machine in the same batch. Each sample was analyzed by GC-MS, UHPLC-MS positive mode and UHPLC-MS negative mode, respectively, to get the best coverage of the metabolites. IF a metabolite was detected simultaneously in GC-MS, UHPLC-MS negative mode and/or UHPLC-MS positive mode, the one with the smallest relative standard deviation (RSD) in the QC samples was retained (lines 156-158).

In GC-MS, the data was normalized with sample weight and the intensity of the internal standard (lines 158-160). In UHPLC-MS, the data was normalized to QC samples and sample weight (161-164). 

a. What about internal/external standards?

Answer: For GC-MS analysis, sorbitol was used as internal standard and the GC-MS data was normalized with sample weight and the intensity of the internal standard, which has been added in the revised MS (lines 114, 158-160).

For UHPLC-MS analysis, a quality control (QC) sample instead of internal standard was used. 1) It is not practical feasible to use internal standards in non-targeted metabolomics, considering different properties of the large number of metabolite features analyzed (Weiwei Wen, et al., 2013, Nature Communications). 2) The addition of internal standards will cause ion suppression of the co-eluted metabolites because the ionization step of UHPLC-MS is a competitive process (Vinayavekhin N and Saghatelian A, 2010, Current Protocols in Molecular Biology). 

b. What about QC samples? How was batch variability handled?

Answer: We are sorry for not mentioning the QC samples in previous version of our manuscript. 

The QC samples were used for both GC-MS and UHPLC-MS analysis. The quality control (QC) sample is a pool of all experimental samples, which was run after every 10 experimental samples (lines 130-131, 142-144). 

All the samples including leaves and seeds were analyzed in one batch in both GC-MS and UHPLC-MS analyses. Therefore, we avoided the batch variability, which was one of the reasons why we only used four replicates for each sample. 

c. How were the samples collected? Was the sample order randomized? Were the GC-MS and UHPLC-MS collected simultaneously? Sequentially? On different days? Was the same sample sub-sampled for GC-MS and UHPLC-MS or were different samples prepared?

Answer: Thank you for clarifying all those important issues we did not mention in the previous version.

The samples of GC-MS and UHPLC-MS analysis were collected and prepared simultaneously. Therefore, the same sample was sub-sampled for GC-MS and UHPLC-MS analysis (lines 106-109). However, the metabolite extraction and mass data collection for GC-MS and UHPLC-MS were done separately (lines 112-126 and lines 133-137, respectively). 

The samples were analyzed in random orders for both GC-MS and UHPLC-MS. The quality control (QC) samples were run after every 10 experimental samples in both GC-MS and UHPLC-MS (lines 130-131, 142-144).

IF a metabolite was detected in GC-MS, UHPLC-MS negative mode and/or UHPLC-MS positive mode, the one with the smallest relative standard deviation (RSD) in the QC samples was retained (lines 156-158).

To make it clear, we have revised M&M section (lines 110-190).

d. Could all of the observed variance be attributed to how the samples were prepared and handled, and how the spectra were collected, instrument variation and insufficient normalization? Because of the nature of MS data, it is “easy” to obtain distinct groups, but it can be difficult to validate that the differences are real.

Answer: Thanks for your concerns. As we mentioned before, our lab had long history of plant metabolomics study and a good practical SOP to reduce as much as possible of the variations occurred during sample collection, preparation, extraction, machine running, data normalization, and ect. 

Because we have already published several articles in different prestigious Journals, in the previous version we neglected the detailed information of all the process, which could be one reason that you have so many technical questions, which we appreciated very much.

We have revised our M&M section (lines 102-182), to provide clear information for your re-judgement.

3. On pages 8-10, the authors over-interpret the significance and the meaning of the relative/comparative meaning of scores/loadings from the various PCA models. Relative trends within a given model are fine, but there is not a point of reference to compare between PCA models. The appearance of a PCA scores plot can change dramatically from any number of reasons, such as changing the order of the samples in the data matrix, changing normalization/scaling methods, removing/adding samples, etc. Given that this section does not provide any real insight or contribute significantly to the study, I would recommend removing it. The fact that the metabolomes vary as a function of time and cultivar is sufficient.

Answer: Thank you again for this question. To avoid over-interpret, we have removed this part in the revised manuscript as suggested.

4. In the data analysis section, the authors state: “False Discovery Rate is chosen for multiple testing correction.” But, it is not clear if all reported p-values are FDR corrected p-values or what type of FDR correction method is used.

Answer: Thanks for this clarification.

First, FDR corrected p-values are only for Two-way ANOVA. 

FDR is based on the Benjamini–Hochberg procedure and significance threshold is defined as the corrected p-value (FDR) < 0.05.

To make it clear, we have revised it in M&M section (lines 173-175). 

5. The heatmap, Figure 4, should be plotted with all biological replicates (not average values) and with hierarchal clustering in both dimensions. This may further emphasize additional consistencies (like Figure 4B) or other trends across the entire dataset.

Answer: Thanks for your suggestion.

Actually, it is a special issue in this case, because the aim of Fig 4 was to present the metabolic alterations in flag leave undergoing senescence. Because of the large variation in metabolite levels in different cultivars, we have to divided the metabolite levels at 7, 14 and 28 DAF with those at 7 DAF, respectively, to eliminate the cultivar-dependent variation. In current Figure 4, the samples were arrayed in such a cultivar and time order that it’s easy to see conserved and divergent change patterns of each metabolite across four cultivars. Such a patter, however, could not be seen if both dimensions were added. In addition, current Fig 4 had already 12 columns, and it would be 48 columns if all biological replicates were presented, in which it’s much harder to discover abovementioned metabolic change patterns. 

However, we have reorganized Fig 4A in the revised manuscript.

6. I understand the choice behind grouping the metabolites by class in the correlation map (Figure 6). But, as is, it fails in being informative. Instead, the metabolites should be ordered from the highest positive to the lowest negative correlation. Then, clusters of metabolites that belong to the same metabolic pathway, cellular process or chemical class within the highly positive/negative correlation should be labeled.

Answer: Thank you for this comment. 

Actually, it is the only choice that we can do currently to make such a correlation map in Fig 6. This map was visualized from a large matrix data (bigger than 200×200) not a simple list. Currently, we don’t know how to order metabolites from the highest positive to the lowest negative correlation and then cluster them. 

In Table S6, all the detailed metabolic correlations between rice seeds and flag leaves, with metabolite names, the classes of the metabolite in seeds and in flag leaves, correlation coefficients and p-values, were listed. It is very easy for those who would like to check which metabolites are highly positively/negatively correlated.

Reviewer #2: 

1. Hu and colleagues address here the understudied subject of the metabolic contribution of the flag leaf (a term that refers to the last growing leaf of rice that is key to the carbon flux into the grain). Flag leaf development is coincident with flowering, thus the properties of this leaf are tightly connected to the final outcome in terms of growth, yield and content of the rice grain. For this reason, for understanding the metabolics events taking place during rice seed development as well as sink-source relationship, it is key to study both flag leaf and grain. The study here used an extensive metabolic platform to investigate the metabolic dynamic changes of flag leaves and grain in 4 rice cultivars (2 indica and 2 japonica subspecies) and to give a broad insight into the influx of metabolites from flag leaf to grain, as well as into the outcome of grain in the light of natural variation. As the study is descriptive in nature, it is raising an important contribution to our current knowledge on this important subject and can path the way for future work: it will be very interesting to follow the secondary metabolites identified here as candidates to be transported from flag leaf to seeds, and their role .

The presentation of the work is easy to follow; data and experiments are well documented and in the data analysis appropriate statistical analysis was applied.

Answer: Many thanks for your support to our work.

I have few minor comments / suggestion:

2. Figure 5: in my opinion, this figure could be shifted to a supplementary file.

Answer: Thanks for your suggestion. However, we decided to keep it in the main text. Fig 5 gave us an overview of the metabolic association between flag leaves and developing seeds, which is an important result for this study. 

3. Page 28 lines 452: pipecolic acid has indeed emerged in recent years as a key defense mediator. I can see the possible role of this signal metabolites in grain for deterring pathogens. Could you also add here a sentence about the possible negative correlation (impact) between Pip and final grain weight as a result of the defense cost.

Answer：Thanks for your suggestion. We did not have the data to show the possible negative correlation between pipecolic acid and grain weight, and we did not find such information in the literature either, therefore, we decided not to make the overstatement on this issue. In addition, we are not sure if the increase of production of pipecolic acid in seeds causes tradeoff of grain weight. Therefore, we did not include this sentence in the revised MS. 

Reviewer #3: 

1. The study of Hu et al titled "Dissection of flag leaf metabolic shifts and their relationship with those occurring simultaneously in developing seed by application of non-targeted metabolomics" investigated the metabolic changes of flag leaves in two japonica and two indica rice cultivars using non-targeted metabolomics approach. This study revealed that flag leaf metabolomes varied significantly on both species and developmental stage, with only a few of the metabolites in flag leaves showing the same pattern of change in the four tested cultivars. The authors also found that levels of 45 metabolites in seeds that are associated with human nutrition and health correlated significantly with their levels in flag leaves.

This study is well organized and provides important insight into the nature of the metabolic flux from source to sink organs in rice.

Answer: Many thanks for your support to our work.

Some minor comments are as follows:

2. Line 34, &135: Please use the uniform presentation of Principal component analysis (PCA) throughout the manuscript.

Answer: Thanks for the suggestion. We have used uniform presentation of principal component analysis (PCA) through the manuscript as suggested.

3. Line 47: “in the word” ?

Answer: Thanks for the correction. We have changed “word” for “world”.

4. Line 176-177: “relatively small, relative large”, I feel the usages are quite strange.

Answer: Thanks for the suggestion. We have rephrased the sentence to “The differences of the interaction scores among different cultivars at 7 and 14 DAF were smaller than those at 0 DAF and 28 DAF.”

5. Line 292-293: Please rewrite this sentence.

Answer: Thanks for the suggestion. We have rewritten this sentence as follows: “Eight of these ten metabolites were phospholipids and the other two of them were N-feruloylputrescine II and uracil (Table 3).”

6. Line 325: rice grain or rice seed? It would be better to keep it uniform throughout the manuscript.

Answer: Thanks for the suggestion. We have used “rice seed” throughout the revised manuscript.

7. Line 351: R-value or r value? It would be better to keep it uniform throughout the manuscript.

Answer: Thanks for the suggestion. We have used “r-value” throughout the revised manuscript.

8. Line 354: the word “arguably” used here is quite confusing.

Answer: Thanks. We have deleted the word “arguably”.

9. Line 446: “that fact that flux modelling in in”, a “in” should be deleted.

Answer: Thanks for the correction. We have deleted a “in”.

10. Line 504: “giving” should be “given”

Answer: Thanks for the correction. We have corrected this error.

11. Line 527: “these species” are ambiguous

Answer: Thanks. We have changed “these species” to “rice”.

Reviewer #4: 

1. Metabolic changes in flag leaves and developing seeds of four rice cultivars were analyzed. Limited overlap in metabolite features were observed among the four lines. Forty-five metabolites enriched in seeds showed a similar pattern of accumulation in leaves and seeds. The authors hypothesize that these data “revealed not only the function of the tissue-specific metabolites but also provide important insight into the nature of the metabolic flux from source to sink organs in rice”. Not certain if this was actually done, there were no flux measurements. Just levels at discrete timepoints, with the leaf samples analyzed much later than the seed samples. There are number of places in the manuscript with long run-on sentences with diverse ideas. These should be rewritten with simpler sentences.

Answer: Thanks for the comments and suggestions. We have changed “metabolic flux” to “metabolites transported” of the sentence.

The leaf samples and seed samples were collected, prepared and extracted at the same time, and their mass spectrometry data was collected at the same time as well (lines 104-111; lines 122-154). The written of the manuscript of the leaf metabolomics was done much later than that of seed samples.

In addition, we have rewritten the long sentences with simpler sentences as suggested (lines 27-30, 73-76, 79-84, 93-95, 97-100, 207-210, 315-318, 388-394, 402-406, 406-410, 443-447, 451-454, 454-458, 473-475, 496-500, 507-509, 511-515).

2. Abstract:

Lines 40-44 needs to be rewritten, especially lines 42-44 which should be deleted. These statements have nothing to do with this study (they should be ok in the last section of the discussion).

Answer: Thanks for the suggestion. We have rewritten this sentence as suggested (lines 22-37).

3. Introduction: Line 57 – brackets are missing for reference 3.

Answer: Thanks for the correction. The brackets have been added in the revised manuscript.

4. Lines 57-62 – Please rewrite, not sure what the authors are trying to convey here.

Answer: Thanks for the suggestion. We have rewritten this sentence to make it clearly understood as suggested (line 55-59). 

5. Line 102 – we report changes in metabolite levels in flag leaves and developing seed from flowering to seed desiccation. The (our) aim was to reveal the source and sink metabolite relationships in rice.

Answer: Thanks for the suggestion. We have rewritten them (lines 97-100).

6. Results: Lines 145 to 147 – these varieties show similar senescence patterns (from Introduction), so it should not be surprising that their metabolomes were relatively similar. Also, for Lines 146 to 149.

Answer: Thanks for the suggestion. According to the suggestion of Reviewer 1, we have deleted this paragraph. 

7. Line 160 – Figure 2 C, the line joining the 2 sets of cultivars should be deleted. They need to describe a bit better why Qingfengai has a very different pattern than the other 3 cultivars. Not fully certain about the need for Figure 2. Figure 1 sums up their data and Figure 3 and later show differences if any among the 4 cultivars.

Answer: Thanks for the suggestion. 

The line joining the two sets of cultivars in the previous Figure 3C has been deleted.

We have added explanations in the revised manuscript to explain why Qingfengai has a very different pattern than the other 3 cultivars (lines 236-239). 

Fig. 2 gave us an additional and deeper view of the result, helping us to decompose the raw data and to explore the causing factors for observed variation. For example, it revealed significant different pattern of Qingfengai than other 3 cultivars and also uncovered the smaller interaction scores among different cultivars at 7 and 14 DAP, which was explained as well (lines 240-243). Therefore, we insisted keep Fig. 2. 

8. Line 237 – should be “extent”

Answer: Thanks for the correction. We have corrected this error.

9. Figure 3 – quite surprising that only linolenate increased with time in flag leaves (expected from lipid breakdown), and the overall patterns for the other metabolites very similar (all decrease over time).

Answer: Actually, there were many metabolites increased with time as linolenate, such as phytocassane C. Please refer to Figure 3 and Table S3. Linolenate is the only increased one in well-modeled metabolites.

10. Line 241-244 – Not sure what this sentence intends to convey? Would not changes in amino acids in flag leaves be largely driven by senescence between 14 and 28 days? Is it not more likely that proteins broken down to amino acids were being mobilized from the flag leaves to developing seeds? The remarkable consistency in data shown in Figure 4B would suggest similarities in the senescence process.

Answer: Thanks very much for these great suggestion. We have revised this paragraph as suggested (lines 316-320). 

Discussion:

A considerable problem the authors face in interpretation is as follows:

11. By and large a bulk of the metabolites unloaded into developing seeds can be expected to be monomers, which are likely to be converted rapidly within developing seeds into polymers, such as proteins, starch, membranes, RNA, and DNA. That is flag leaf metabolism is dissimilatory during senescence, whereas seed metabolism is assimilatory prior to desiccation. The authors consider this point but should discuss the shortcomings of their current experimental protocol in somewhat greater detail.

Answer: Thanks for the suggestion. We have added a paragraph named “Prospect for future research” at the end of the discussion section as you suggested (lines 566-578).

12. A case in point is Table 4, where they detected significant associations in metabolites between seeds and leaves. However, it is difficult to discern if there was/is a direct biological correlation between these levels in the two tissues. Can they be sure these compounds came from the flag leaves to seeds, and say not from root transfer?

Answer: Thanks for the suggestion. 

We have rephrased this sentence to clarify this issue (lines 552-557). We cannot exclude the possibility that part of them come from roots.

13. There are other aspects the authors should consider – both flag leaf senescence and seed development in rice have been intensely studied using physiological, biochemical and molecular tools. It may help if they could incorporate key findings from such studies within the scope of their metabolite analyses. They mention as much in their conclusions.

Answer: Thanks for the suggestion. 

We have revised the manuscript and incorporated some metabolomics findings with previous findings from other omics (lines 478-479, 515-5519, 530-532, 552-557).

14. It might be best to revise the manuscript to stay within the key elements of their work (flag leaf metabolomes) and draw a simpler inference to their earlier work with seeds [ref 20]. It is also not clear if the instrument settings and detector sensitivity were the same for the two experiments. There was no mention of an internal standard used if it was they should add this to their M&M section.

Answer: Thanks for the suggestion. Actually, it would be simpler if we just revised the manuscript to stay within the flag leaf metabolome. However, it would be too simple, because we did not collect other leaves. Because all leave samples and seed samples were collected, prepared, extracted and analyzed exactly at the same time, it would be a pity that we would not correlated them together, although it proved to be a hard task due mainly to the lack of metabolic flux analysis. 

For your questions regarding the instrument settings and detector sensitivity, please refer the answers to the #3 question of the Reviewer #1, or to the lines 336-339. 

An internal standard was used for GC-MS analysis, which have been added in M&M section (lines 114 and 160)

Materials and Methods:

15. Lines 482-484 appear to be repeated?

Answer: Thanks for the correction. We have revised as suggested.

16. I am not sure I fully understand their approach to metabolites missing in one or more samples. It might be best to exclude these metabolites from further analyses. It was either present or absent in a given extract. What exactly is the median value of a metabolite, area in all four samples, area in 2 out of 4 replicates? Please clarify. The authors could consider treating them as missing values and make comparisons based on samples or treatment groups with detectable levels. Another alternative would be to do a binomial test (e.g. Chi-square) based on presence or absence of a metabolite.

Answer: Thanks.

For non-targeted metabolomics, it is very common to see missing value in one or more samples, especially when natural variation is large. In this study, we carefully checked each metabolite in each sample with Mass Profinder software to make sure that missing values are caused by the content being too low to be detected but not by random (lines 152-154). Thus, the metabolites with missing values were also retained for further statistical analysis. 

There were many missing value imputation approaches for mass spectrometry-base metabolomics data analysis (Wei et al., 2018, Scientific Reports). We accepted general rule to impute missing values with the detected minimum value of the same metabolites in other samples for statistical analysis.

We have rephrased the method for missing value imputation and data normalization in the revised manuscript (lines 164-166).

---

## [Editor Report · Decision Letter 1]

23 Dec 2019

Dissection of flag leaf metabolic shifts and their relationship with those occurring simultaneously in developing seed by application of non-targeted metabolomics

PONE-D-19-26924R1

Dear Dr. Shi,

We are pleased to inform you that your manuscript has been judged scientifically suitable for publication and will be formally accepted for publication once it complies with all outstanding technical requirements.

With kind regards,

Haitao Shi

Academic Editor

PLOS ONE
---

## [Editor Report · Acceptance letter]

30 Dec 2019

PONE-D-19-26924R1 

Dissection of flag leaf metabolic shifts and their relationship with those occurring simultaneously in developing seed by application of non-targeted metabolomics 

Dear Dr. Shi:

I am pleased to inform you that your manuscript has been deemed suitable for publication in PLOS ONE. Congratulations! Your manuscript is now with our production department. 

With kind regards,

on behalf of

Dr. Haitao Shi 

Academic Editor

PLOS ONE